

# An artificial-vision- and statistical-learning-based method for studying the biodegradation of type I collagen scaffolds in bone regeneration systems

Yaroslava Robles-Bykbaev[1,2,3], Salvador Naya[4], Silvia Díaz-Prado[1],
Daniel Calle-López[2], Vladimir Robles-Bykbaev[2], Luis Garzón[3], Clara
Sanjurjo-Rodríguez[1] and Javier Tarrío-Saavedra[4]

[1] Instituto de Investigación Biomédica de A Coruña (INIBIC), Complexo Hospitalario Universitario de A Coruña (CHUAC), SERGAS, Departamento de Medicina, Universidade da Coruña, A Coruña, Spain
[2] Cátedra UNESCO UPS Tecnologías de apoyo para la Inclusión Educativa, Universidad Politécnica Salesiana, Cuenca, Ecuador
[3] Grupo de Investigación en Materiales (GiMaT), Universidad Politécnica Salesiana, Cuenca, Ecuador
[4] Grupo MODES, CITIC, ITMATI, Departamento de Matemáticas, Universidade da Coruña, Ferrol, Spain

Corresponding author
Javier Tarrío-Saavedra,
javier.tarrio@udc.es

## ABSTRACT

This work proposes a method based on image analysis and machine and statistical learning to model and estimate osteocyte growth (in type I collagen scaffolds for bone regeneration systems) and the collagen degradation degree due to cellular growth. To achieve these aims, the mass of collagen -subjected to the action of osteocyte growth and differentiation from stem cells- was measured on 3 days during each of 2 months, under conditions simulating a tissue in the human body. In addition, optical microscopy was applied to obtain information about cellular growth, cellular differentiation, and collagen degradation. Our first contribution consists of the application of a supervised classification random forest algorithm to image texture features (the structure tensor and entropy) for estimating the different regions of interest in an image obtained by optical microscopy: the extracellular matrix, collagen, and image background, and nuclei. Then, extracellular-matrix and collagen regions of interest were determined by the extraction of features related to the progression of the cellular growth and collagen degradation (e.g., mean area of objects and the mode of an intensity histogram). Finally, these critical features were statistically modeled depending on time via nonparametric and parametric linear and nonlinear models such as those based on logistic functions. Namely, the parametric logistic mixture models provided a way to identify and model the degradation due to biological activity by estimating the corresponding proportion of mass loss. The relation between osteocyte growth and differentiation from stem cells, on the one hand, and collagen degradation, on the other hand, was determined too and modeled through analysis of image objects' circularity and area, in addition to collagen mass loss. This set of imaging techniques, machine learning procedures, and statistical tools allowed us to characterize and parameterize type I collagen biodegradation when collagen acts as a scaffold in bone regeneration tasks. Namely, the parametric logistic mixture models provided a way to identify and model the degradation due to biological activity and thus to estimate the corresponding proportion of mass loss. Moreover, the

proposed methodology can help to estimate the degradation degree of scaffolds from the information obtained by optical microscopy.

## INTRODUCTION

Currently, the analysis of images of cell growth and differentiation from one type of lineage to another is to a great extent qualitative (*Basiji et al., 2007*). This method of analysis, which is based on observation of images, does not yield robust quantification of the changes produced during the cell growth and differentiation.

The qualitative analysis is currently performed on images obtained by biomedical imaging techniques such as transmission electron microscopy and scanning electron microscopy. This analysis is often based on examining (in a subjective way) the electrodensity of related objects that appear in the obtained images (cells, cell nuclei, cytoplasm, biomaterials such as type I collagen, and extracellular material, among others) (*Sanjurjo-Rodríguez et al., 2017*; *Sanjurjo-Rodríguez et al., 2014*; *Sanjurjo-Rodríguez et al., 2016*; *Sanjurjo-Rodríguez et al., 2013*; *Martínez-Sánchez et al., 2013*; *Gashti et al., 2012*). Therefore, the information obtained with this type of microanalytical techniques is largely of a qualitative nature (elements observed in a certain area of the image are analyzed by specialized personnel) (*Gashti et al., 2012*). Specifically, such studies determine how electrodense the particles and/or areas contained in the image are (*Sanjurjo-Rodríguez et al., 2014*). Implementation of a quantitative analysis is also necessary, to evaluate the observable changes in a more general, reproducible, and reliable way (*Wootton et al., 1995*; *Appel et al., 2013*; *Ong, Jin & Sinniah, 1996*; *Tayebi et al., 2012*; *Teverovskiy et al., 2004*; *Han et al., 2012*; *Appel et al., 2013*; *Tabesh et al., 2007*), independently of the imaging technique that is used (e.g., ultrasonography, photoacoustic microscopy, magnetic resonance imaging, optical imaging, X-ray imaging, and nuclear magnetic resonance imaging) (*Appel et al., 2013*; *Nam et al., 2014*; *Tayebi et al., 2012*). A method based solely on the criterion of the observer, even an expert observer, may be not able to quantitatively verify the level of reliability of the information present in the image. This situation poses a risk of wrong and skewed decisions and conclusions based on the analyzed results. In addition, these tasks are often difficult (requiring skilled personnel) and highly time consuming. Thus, the implementation of image segmentation techniques and statistical analysis of the image information that automatize this process are necessary (*Han et al., 2012*; *Appel et al., 2013*; *Tarrío-Saavedra et al., 2017*).

Nowadays, there are techniques related to image analysis that allow researchers to overcome the limitations of qualitative analysis. In this field, segmentation of images is a fundamental technique for identification of objects, such as cells in tissues. Among the methodologies of image segmentation, there are the thresholding method, clustering

approach, region-based approaches, edge detection approaches, and others (*Patil & Deore, 2013*).

The techniques based on artificial intelligence also allow for automation of processes and for improving decision making based on the components and objects of each image. Thus, this approach constitutes a new image-diagnostic technology capable of quantifying the changes related to the biological activity shown in images. That is, the techniques of image analysis become scientific support during analysis of images of cellular tissues and their components. The imaging techniques facilitate the task of technical specialists when they deal with the analysis of observable biological changes in the images of cells in tissues.

## Image segmentation

Within this framework, image segmentation techniques provide information about the number of cells and size, shape, and extension of the tissue in each micrograph. The study of the number, shape, and extension of the cells as a function of time provides valuable information about the degree and rate of degradation of scaffolds such as collagen for the culture of bone cells. Application of statistical regression models to these variables can extract information about the mechanisms of biological degradation of the material, thus predicting the level of degradation due to biological activity and, as a result, making decisions and creating selection criteria (*Tarrío-Saavedra et al., 2017*; *Janeiro-Arocas et al., 2016*).

In the research on tissues, the image analysis is a complex task that requires—in addition to broad knowledge of analytical techniques—deep knowledge about the problem at hand (*Teverovskiy et al., 2004*; *Suvarna, Layton & Bancroft, 2018*; *Bozzola & Russell, 1999*). Namely, tasks such as differentiation between an extracellular matrix and scaffold (biomaterial) require well -trained and experienced personnel (*El-Jawhari et al., 2016*; *Sanjurjo-Rodríguez et al., 2016*). Modern imaging techniques such as those based on optical and digital microscopy, scanning electron microscopy , and flow cytometry, among others, involve a long process for which the knowledgeability of technicians and researchers is crucial (*Appel et al., 2013*; *Nam et al., 2014*; *Gashti et al., 2012*; *Tayebi et al., 2012*). In addition, the analysis performed to identify the processes in which certain types of cells grow and differentiate into other lineages is generally based on the subjective opinion of the analyst. Therefore , as indicated by *Basiji et al. (2007)*, there is a lack of morphological studies that support these analyses. In fact, as *Meijering (2012)* suggests, the main problem of image analysis is image segmentation, one reason being that the type and quality of the acquired images have a strong influence on the success of cell segmentation (correct identification of objects) (*Kasprowicz, Suman & OToole, 2017*). On the other hand, the choice of an appropriate segmentation procedure and its parameters also depends on the work of personnel specialized in the concrete problem that needs to be solved.

Among all the techniques of image segmentation, there are several approaches such as thresholding and its adaptive variants (*Zhao et al., 2014*; *Cheng et al., 2013*), approximations based on clusters (*Wang & Pan, 2014*; *Gong et al., 2013*), and certain mixed models (*Cheng et al., 2013*), and other alternatives.

## Statistical modeling

The statistical techniques applied to the extracted vector of features after segmentation are another key topic in this domain. Now, the statistical analysis performed on two-dimensional (2D) and 3D images is often based on methods mainly focused on a descriptive analysis of the data. This exploratory analysis often includes calculation based on measurements of a position (mean) and dispersion (standard deviation) in addition to unsupervised classification methodologies for identifying groups of objects (e.g., cells, membranes, extracellular material, and scaffold biomaterial) segmented in the images. At a population level, the exploratory data analysis does not provide information about the relation between cell growth and differentiation from stem cells or about degradation of the bioscaffold. Therefore, there is a lack of application of inferential statistical models. These could identify and explain possible correlations between, on the one hand, predictors that account for the cell growth and differentiation and, on the other hand, the degradation level of biomaterials.

The statistical study of scaffold degradation under the action of cell growth is necessary to choose a more adequate biomaterial for each application. In fact, there are works involving applied mathematics and statistical techniques that estimate the degradation level of biomaterials depending on critical-for-degradation variables such as the type of biomaterial, cell group, culture medium, cell growth duration, and time of degradation of the biomaterial (*Chen, Zhou & Li, 2011*; *Hoque, Yong & Ian, 2012*; *Pitt & Zhong-wei, 1987*; *Sandino, Planell & Lacroix, 2008*). *Chen, Zhou & Li (2011)* developed a numerical model taking into account stochastic hydrolysis and mass transport to simulate a process of degradation of biomaterials and their erosion. *Hoque, Yong & Ian (2012)* modeled the mass loss using an exponential expression, assuming that the diffusion of water and hydrolysis are the main causes of the biomaterial degradation processes. For our purposes, tools of statistical learning (the field dealing with the interrelation of statistics and computing) for complex data have been applied to model the trends of degradation corresponding to the materials being analyzed (*Friedman, Hastie & Tibshirani, 2008*). The aforementioned studies show the need for statistical modeling of the mechanical, physical, and rheological phenomena associated with biomaterials such as type I collagen as well as the modeling of degradation to which these biomaterials are subject in cell culture (in addition to the growth and cellular differentiation modeling).

In the present work, the data obtained from the segmentation process were analyzed by applying nonparametric and parametric regression models to determine the degree of degradation of type I collagen. The adjusted models were evaluated and compared using criteria of goodness of fit such as the coefficient of determination, $R^2$, according to the definition proposed by *Hayfield & Racine (2008)*.

The computational tool that was used for the model estimation and evaluation is the R statistical software (*R Core Team, 2017*). Specifically, packages `nls2`, `grofit`, and `minpack` were employed to fit nonlinear regression models, whereas the `investr` package (*Greenwell & Kabban, 2014*) were used to obtain an estimate of prediction intervals for the fitted model. In addition, an evolutionary algorithm for global optimization was applied via the `DEoptim` package to obtain an initial solution for parameters of the adjusted nonlinear

models (*Ríos-Fachal et al., 2014*). Alternatively, the `grofit` library could be utilized. It allowed us to fit parametric and nonparametric regression models from an initial solution for the model parameters obtained by means of locally weighted polynomial regression models (LOWESS). Furthermore, the `mgcv` library was used to evaluate generalized additive models (GAMs) based on the fitting of a basis of penalized regression splines (*Wood, 2006*), whereas packages `ggplot2`, (*Wickham, 2009*), and `RColorBrewer` were utilized to implement graphics.

## The scheme of the proposed methodology

The main contributions of this work to bioscaffold degradation analysis are summarized below. All of them will be conveniently described throughout this document:

- The proposed automatic procedure, based on the random forest supervised classification, to identify different regions of interest (ROIs) in the images obtained by optical microscopy. It is important to emphasize that the random forest classification model is trained taking into account the information provided by qualified laboratory personnel in a qualitative study of this type of images. Therefore, we present a model that is capable of automating a complex and time-consuming task in the context of this type of analysis.
- Once the ROIs were identified, i.e., the extracellular matrix, collagen, cellular nuclei, and background, the present methodology allowed us to separately study their evolution.
- The proposed methodology performs extraction of features related to cell growth in the ROIs of each image.
- Statistical learning methods such as regression model fitting were employed to explain the evolution of relevant characteristics (of cell growth) and of biodegradation of collagen.
- The growth of cells can be characterized by statistical modeling of relevant features such as the total area of cells. These models can help to determine the rate of cell growth, predict cell growth evolution, and to evaluate the use of support materials (in this case collagen) as scaffolds.
- Modeling and estimating the relation between the features of cell growth and of differentiation (from stem cells) and the collagen mass loss (the index of material biodegradation level) were performed too.
- A method for estimating the degree of degradation (collagen mass loss) from the information in micrographs is proposed as well.

## MATERIALS AND METHODS

In this work, we studied the degradation of a biomaterial, namely type I collagen, as a result of cellular activity. Biomaterials are considered mechanically functional and physiologically acceptable products used to safely replicate the function of living tissues in biological systems and are implanted temporarily or permanently into the body. The goal is to restore an existing function and, in some cases, to regenerate tissues (*Alfaro & Fernández, 2011*). Within this framework, because of high biocompatibility, type I collagen

is regarded as the gold standard in tissue engineering (*Silvipriya et al., 2015*) owing to its biocompatibility. Collagen is a biomaterial (the most common protein in the body) that can act as a scaffold for regeneration of bone and cartilage tissues, among other applications in tissue engineering, e.g., injectable matrices and scaffolds intended for bone regeneration (*Sanjurjo-Rodríguez et al., 2017*; *Geiger, Li & Friess, 2003*; *Behring et al., 2008*; *Inzana et al., 2014*; *O'brien, 2011*; *Silvipriya et al., 2015*). Indeed, it has a high potential for cultivation of cells for producing bone because, among other reasons, it is one of the two main components of bone (along with hydroxyapatite), accounting for 89% of its organic matrix and 32% of its volumetric composition (*O'brien, 2011*). To improve its performance as a scaffold, collagen can be employed in combination with other types of substrates such as synthetic materials or bovine bone (*El-Jawhari et al., 2016*).

Human mesenchymal stem cells 3A6 (CMMh-3A6), a line of immortalized mesenchymal stem cells, were provided by Stem Cell, Department of Medical Research & Education and Orthopedics & Traumatology, Veterans General Hospital, Taipei, Taiwan. A total of 4,200,000 CMMh-3A6 cells of passage $\times^+$ were cultured. Two changes of the culture medium and one subculturing (passage 2 or reseeding) were carried out per week because high confluence (90%) can be reached within a relatively short period (approximately between 3 and 4 days) in the plates of an inverted microscope.

The biomaterial (type I collagen, commercial product) was prepared to obtain disc-shaped cuts (with a diameter of 8 mm) performed using a biopsy punch. Type I collagen sponges were arranged in three groups, each consisting of 21 samples. For technical reasons, we call groups with the labels CCO, CCT, and CO groups #1, #2, and #3 , respectively. The 21 samples in group 1 (CCO) were made up of type I collagen, CMMh-3A6 cells, and a commercial osteogenic culture medium, namely the hMSC Osteogenic Differentiation BulletKit$^{TM}$ Medium (Lonza, Spain). Group #2 (CCT) was composed of 21 samples consisting of Dulbecco's modified Eagle's medium (DMEM) supplemented with 1 g/L D-glucose and pyruvate (Gibco, USA), 5% of GlutaMax (Gibco), and 10% of fetal bovine serum (Gibco) as well as CMMh-3A6 cells seeded on the biomaterial (type I collagen). Finally, group #3 (CO) was composed of 21 samples consisting of type I collagen and the commercial osteogenic culture medium (hMSC Osteogenic Differentiation BulletKit$^{TM}$ Medium; Lonza), but without cells. That is, the third group was the control group.

The experiment was conducted for 44 days, under conditions that mimic human organism conditions (in a culture oven): pH = 7.4, temperature = 37 °C, and 5% $CO_2$. Then, histomorphological analysis was performed on the samples, and they were embedded in paraffin, deparaffinized, and stained with hematoxylin and eosin. After that, 2D images were captured by optical microscopy at 4 × magnification. Overall, 60 micrographs were obtained, which were segmented by machine learning algorithms for each analyzed group, thus resulting in binary images. The analyzed parameters that were taken into account for the classification were type I collagen, the extracellular matrix, nuclei, and image background.

It is necessary to extract relevant features from images to train the computational system. These attributes constitute the basis for the learning process. Then, a set of binarized images

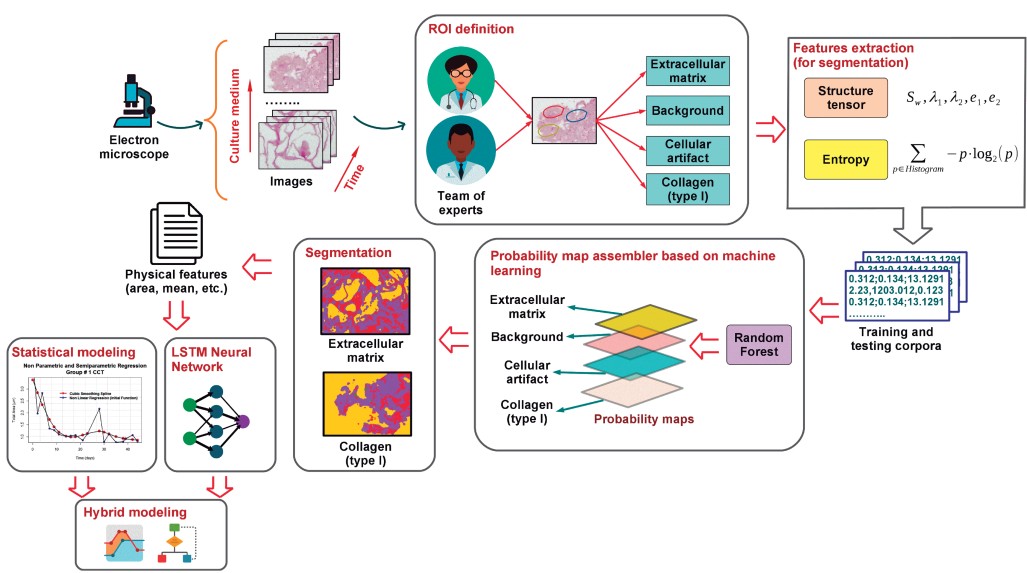

**Figure 1** **A general outline of the proposed modeling of cellular growth and the degree of degradation of type I collagen.**

is extracted (one for each segmentable attribute), from which relevant information for the calculation of these features is also extracted.

## Proposed methodology

This work focuses on the modeling of the degree of degradation of type I collagen and cell growth by segmentation analysis (identification and separation of objects) of images. These images correspond to cultures of bone cells and are obtained by optical microscopy, after staining of the cells with hematoxylin and eosin. The process map in Fig. 1 shows the proposed methodology that was developed here to characterize and model collagen degradation and cell growth. It includes several stages that are roughly outlined below.

To study the biodegradation of collagen as a scaffold for regeneration of bone tissue, CMMh-3A6 stem cells were seeded to differentiate them into osteocytes. Then, images were captured by optical microscopy after staining with hematoxylin and eosin. The images were obtained as a time series within short periods, 3 to 4 days.

Based on the expertise of a team of researchers in the biomedical area, an ROI was defined on which it was desirable to perform the analysis via the extraction of various characteristics (e.g., the area, number of particles, and ratios). These are the extracellular matrix, background, nuclei, and type I collagen.

Once the ROIs were defined, the feature extraction from the images was carried out. These features contain information about the shape, size, and extension of the cells, as related to the progression of collagen degradation. At this stage, the tensor structure and entropy (*Sahoo, Soltani & Wong, 1988*) corresponding to the pixels related to different ROIs of each image were obtained. With these attributes, a training sample was built, when

we knew the actual type of ROI corresponding to each pixel. This sample was used later to perform the segmentation of the images.

After obtaining the training sample, we proceed to apply supervised classification algorithms based on machine learning such as the random forest classifiers (*Breiman, 2001*). To achieve this goal, first, the classes of ROIs were determined. Via the identification of these classes, we estimated a model on the basis of the training sample by associating the ROI of each pixel with its corresponding feature vector (composed of entropy and energy values). Next , the models (based on random forest) assigned the ROI class (resulting from generation of probability maps (*Arganda-Carreras et al., 2017*)) to each pixel of the test sample (the sample in which we did not know the actual ROI).

Once the different classes were identified within each image, and the probability maps were built, we proceed to extract the representative characteristics related to the cellular activity. These features are the area, major and minor axes of a fitted ellipse, circularity, and histogram intensity mode, among other features corresponding to the objects of each image. The extracted characteristics were selected taking into consideration that they provide relevant information about the cellular activity and, by extension, biodegradation of the support material or scaffolding, in this case, type I collagen. Software tools ImageJ (*Schneider, Rasband & Eliceiri, 2012*; *Schindelin et al., 2015*) and Waikato Environment for Knowledge Analysis (Weka) (*Hall et al., 2009*) were used to perform the segmentation, classification, and feature extraction tasks. On the one hand, ImageJ was employed to extract the image attributes in order to apply thresholding processes and to generate the corpus of data related to each ROI. On the other hand, Weka was applied to training of the random forest classifier that provided information about the identification and categorization of different ROIs.

Finally, a statistical learning-model approach based on parametric (linear and nonlinear) and nonparametric regression models was applied to estimate the value of relevant characteristics of the biological activity extracted as a function of time. These estimates are an indicator of the degree of collagen degradation. These estimated models provide information about the type and degree of dependence between variables. They also offer prediction of the degradation degree from image features. More information about texture analysis of original micrographs, the segmentation process via the random forest classifier, and application of statistical model ing of the scaffold degradation degree is provided in the text below.

### Texture analysis

The approach proposed in this study is to quantitatively determine cell growth and differentiation and the degree of degradation of type I collagen due to a biological activity. The first step for achieving these objectives is to associate a numerical vector of features with each pixel. In fact, once the different ROIs were identified by expert personnel, readers can see that the different regions can be differentiated according to the texture of the image. For this purpose, scale-invariant texture filters were applied to 2D images by means of the ImageJ software. Image texture can be defined as a spatial arrangement of the color or intensities of an image in a given region, whereas texture analysis is often performed

to separate different regions of an image. Therefore, in this paper, we propose to apply an image texture analysis consisting of extraction of representative features, such as a structure tensor and entropy, to identify different ROIs.

Once the characteristic vectors are extracted, they will serve as a training sample for the classifier, in this case, the random forest algorithm.

1. Structure tensor: This is a matrix representation of the image's partial derivatives defined as second-order symmetric positive matrix J:

$$J = \begin{bmatrix} \langle f_x, f_x \rangle w & \langle f_x, f_y \rangle w \\ \langle f_x, f_y \rangle w & \langle f_y, f_y \rangle w \end{bmatrix} \tag{1}$$

where $f_x$ and $f_y$ are images of the partial spatial derivatives: $\frac{\partial f}{\partial x}$ and $\frac{\partial f}{\partial y}$, respectively (*Budde & Frank, 2012*). From this matrix, all major and minor eigenvalues are separated for each pixel and channel in the image:

$$T(v) = \lambda v \tag{2}$$

where $\lambda$ is the eigen values, and **v** denotes the eigen vectors.

2. Entropy: For its calculation, a circle of radius $r$ is drawn around each pixel. The image intensity histogram corresponding to the separated circle is obtained as binarized image fragments. Finally, entropy is calculated for each particle, where $p$ is the probability of each chunk in the histogram corresponding to each channel of the image, both in RGB and in Hue, Saturation Brightness formats.

### The random forest classifier for identifying different ROIs

It is possible to apply a wide variety of supervised classification models based on the descriptors that were extracted from the images in order to identify different ROIs. For the specific case of this research, a random-forest–type classifier (*Criminisi, Shotton & Konukoglu, 2012*) was chosen. It has the advantages of operating on attributes (or modular characteristics), of preventing overfitting of certain classes, and an optimized computational cost.

In recent years, decision trees proved to be some of the most promising techniques in machine learning, computer vision, and medical analysis (*Criminisi, Shotton & Konukoglu, 2012*). Random forest classifiers operate by constructing several decision trees (predictive processes that map observations of an item to conclusions about the objective value of the item) in the training phase, to then produce a class fashion (by its nature as a classifier) for each tree.

### Image segmentation

Now, the benefits of the technology based on artificial vision allow us to determine the biological behavior of cells such as cell growth and differentiation as well as degradation of biomaterials such as type I collagen. Image segmentation is a reference within the support techniques for achieving this goal. In general, image segmentation is defined as the process of dividing an image into different segments or groups of pixels that share certain common characteristics (*Mallik et al., 2011*; *Janeiro-Arocas et al., 2016*). This is a fundamental task in image analysis for the detection of objects (*Mallik et al., 2011*; *Janeiro-Arocas et al., 2016*). In fact, it is a central problem in many studies dealing with image analysis (*Meijering, 2012*). It

is important to note that the type and quality of the acquired images influence the success of cell segmentation (identification and separation of objects) (*Kasprowicz, Suman & OToole, 2017*). Therefore , adequate quality of images is one of the existing requirements of this set of techniques. In fact, an important feature of image segmentation is that although it is conceptually simple, it lacks generality; consequently, it cannot be implemented reliably and easily for all cell lines, image modalities, and cell densities without preprocessing of images (*Alanazi et al., 2017*).

Limiting factors for the image segmentation process are, e.g., the type of objects to be analyzed in the image, the aims pursued by the research team, and limitations of the knowledge of the technician in charge of the segmentation process. All these factors necessitate devising specific strategies for the processing and analysis of images for each particular problem. Despite the absence of a universal segmentation procedure (*Alanazi et al., 2017*), today it is possible to analyze 2D images of the behavior of stem cells in vivo, with information on sequential growth and differentiation as a function of time, via a set of different segmentation techniques. Among the most popular segmentation and image-processing techniques are thresholding, region-growing methods, edge detection, and Markov random field color algorithms (*Grys et al., 2017*). In this work, we propose a segmentation method based on a random forest classifier trained on vectors of attributes and on properties obtained from filters that are applied to labeled ROIs in the images. In fact, different studies have revealed the great potential of the random forest method as a support tool in the segmentation process. Thus, *Schroff, Criminisi & Zisserman (2008)* demonstrated the potential of random forest as a technique for segmentation processes of objects present in images. They worked with the MSRC image dataset, incorporating both features that describe objects locally and those that characterize the environment of those objects. In the medical domain, *Dhungel, Carneiro & Bradley (2015)* identified formation of unusual masses in mammograms. These masses varied in size, texture, and the area that delimits them, merging with the rest of the tissues of the breast. The classifiers based on deep belief networks (first filtering in which suspicious areas are located) and random forest (last filtering) were applied in a cascade. This approach significantly reduced the number of false positives. Furthermore, *Khan, Hanbury & Stoettinger (2010)* proved that the random forest technique is highly effective at segmenting human skin images. Those authors used random forest in combination with Improved Hue, Luminance and Saturation color space and compared this classifier with others based on Bayesian networks, multilayer perceptron, support vector machine, AdaBoost, naïve Bayes, and Radial Basis Function neural networks. The classifier surpassed all other techniques, yielding 0.877, 0.738, and 0.740 in terms of accuracy, precision, and recall, respectively. Finally, *Ghose et al. (2012)* demonstrated the effectiveness of segmenting magnetic resonance images of human prostates. To this end, those authors propose to use a framework of decision trees in order to obtain a probabilistic representation of those voxels that define the prostate.

The supervised classification process was applied by means of the Weka software (*Hall et al., 2009*). This computational tool allows for the application of a wide range of techniques from the field of data mining, e.g., data visualization techniques, feature

selection, unsupervised classification or clustering algorithms, supervised classification such as random forest, and regression models (*Frank et al., 2004*).

### Regression model fitting

Once the vectors of features related to the growth and cellular differentiation of osteocytes were obtained through the process of image segmentation (mentioned above), the degree of degradation of type I collagen was estimated via application of statistical regression models. In addition, it is important to note that the extracellular matrix ROI is extracted from the image and its features. They are the area, objects' circularity ($4\pi \, area/perimeter^2$), and the intensity histogram mode, among others that are modeled as a function of time to estimate cellular growth. The relation between collagen mass loss and the area or circularity was studied here to model the degree of degradation of collagen as well as the biological activity.

Two main regression approaches were employed: nonparametric and parametric nonlinear regression. In the text below, the two aforementioned approaches are briefly introduced.

Nonparametric methods have attracted the attention of academia and industry (*Hayfield & Racine, 2008*). They have relevant advantages, e.g., they do not assume any parametric function that constrains the relation between variables or even a specific distribution of those variables of interest (*Tarrío-Saavedra et al., 2011*). Indeed, nonparametric methods provide estimates of flexible models that do not impose any prespecified function (*Maity, 2017*), thereby allowing investigators to model a wide range of possible complex nonlinear functions (*Shokrzadeh, Jozani & Bibeau, 2014*). There are many types of nonparametric models dedicated to regression tasks, for example, kernel smoothing (*Wand & Jones, 1994*), local polynomial regression (*Breidt & Opsomer, 2000*), regression splines (*Wood, 2006*; *Mammen & Van de Geer, 1997*), and LOWESS (*Cleveland & Devlin, 1988*).

Specifically, penalized regression spline modeling within the framework of nonparametric GAMs (*Wood, 2006*) is proposed here to obtain information about the type of dependence between collagen mass loss and time. For the sake of simplicity, a penalized B-spline basis was fitted. A generic GAM as a function of two linear predictor variables ($X_1$ and $X_2$) and two smooth predictors ($T_1$ and $T_2$) can be defined as follows:

$$Y = \beta_0 + \beta_1 X_1 + \beta_2 X_2 + s_1(T_1) + s_2(T_2) + s_{12}(T_1, T_2) + \epsilon \tag{3}$$

where the effects on the response $Y$ of $X_1$ and $X_2$ are linear, whereas the effects of $T_1$ and $T_2$ are only assumed to be smooth. This model additionally allowed us to include linear or smooth effects for the interaction between variables, as is the case for $S(T_1, T_2)$. The discrepancy between $Y$ and $Y$ estimates, $\hat{Y}$, was measured by the model residuals, $\epsilon$. When the $Y$ variable is discrete, e.g., binomially or Poisson distributed, $Y$ is replaced by $g(\eta)$. In the present case, $Y = Mass \, Loss$ and $X = Time$; thus,

$$\hat{Y} = E(Mass \, loss | Time) = \beta_0 + s_{Time}(Time) \tag{4}$$

As mentioned above, package `mgcv` provides computational tools to fit GAMs. This library uses penalized regression splines for the estimation of smooth effect $s$ of

each predictor variable $x$. To define the concept of splines, consider the set of nodes $a < t_1 < t_2 < \cdots < t_L < b$ in interval $[a, b]$. A spline function is a polynomial function in each subinterval of a certain degree (e.g., three in the case of a cubic spline), which is continuous in the knots up to order degree $L - 1$. With the aim of determining a spline function, we must know $L + degree + 1$ values, known as the degrees of freedom of the spline family defined for the set of knots. The $degree + 1$ number is known as spline order.

Thus, we can write the smooth effect as

$$s_j(x) = \sum_{k=1}^{L+degree} \beta_{jk} \phi_{jk}(x) = \beta_j' \phi_j \qquad (5)$$

for the basis composed of $L + degree$ functions $\phi_{jk}$ that define the B-splines, where $L$ is the number of knots (10 by default) and $degree = 3$ (cubic regression splines). Given the $s_j$ definition, $\hat{Y} = E(Mass\ loss|Time)$ is now a parametric mean function with parameters $\beta = (\beta_0, \beta_{Time}')'$ and predictor $Time$ that define the intercept and the splines that define $s_j$.

The penalized least squares objective function for estimating $\beta$ is

$$\|Y - X\beta\|^2 + \sum_j \lambda_j \beta_j' B_j \beta_j \qquad (6)$$

The values of $\lambda_j$ are selected by an iterative algorithm to minimize a generalized cross-validation criterion.

Parametric models provide the transfer function that determines the dependence relation between critical variables of a process. In addition, the constitutive parameters usually have a physical meaning. Thus, the chemical, physical, and biological processes of materials and substances can be characterized and compared by studying the parameter values.

To illustrate a parametric-model expression in a simple way, we can define

$$Y = m(t_i) + \varepsilon_i, \qquad \text{with } i = 1, 2, \ldots, n, \qquad \text{and} \qquad E(\varepsilon_i) = 0 \qquad (7)$$

where $Y$ is the response variable to be modeled, and $m$ is the regression function fitted to $Y$. The latter can be the mass loss, area, and circularity, among other features critical for collagen degradation and cell growth. Finally, $\varepsilon_i$ are the independent, identically distributed model residuals. In a fixed design, $0 \leq t_1 \leq t_2 \leq \ldots \leq t_n$, the parametric functions can be defined as $\mathcal{M} = \{m_\theta \ / \ \theta \in \ominus\}$, where $\theta$ is the parameter vector that defines the regression model, and $\ominus$ is a subset of $\mathbb{R}^k$. The linear models are the simplest and most typical parametric models, but in chemistry and biology domains, nonlinear models are also important and useful. In this regard, there are many processes characterized by exponential and sigmoid types of relations between critical variables such as material degradation (*Robles-Bykbaev et al., 2018*; *Ríos-Fachal et al., 2014*), crystallization (*López-Beceiro, Gracia-Fernández & Artiaga, 2013*), oxidation (*Tarrío-Saavedra et al., 2013*), and material dimensional variation due to magnetic changes (*Tarrío-Saavedra et al., 2017*). In fact, the requirements of biological growth have popularized sigmoid monotonic nonlinear functions, characterized by exponential growth and a point of inflection from which the speed of growth gradually decreases until reaching a saturation zone (*Kahm et al., 2010*; *Román-Román & Torres-Ruiz, 2012*; *Kingsland, 1985*; *Tarrío-Saavedra et al., 2014*). Some of the most popular functions are those based on logistic and Gompertz functions:

1. Gompertz function:

$$y(t) = A \exp\left[-\exp\left(\frac{\mu e}{A}(\lambda - t) + 1\right)\right] \tag{8}$$

2. Logistic function:

$$y(t) = \frac{A}{1 + \exp\left(\frac{\mu - t}{scal}\right)} \tag{9}$$

where $A$ is the asymptote, $\mu$ represents the inflection point or time corresponding to the maximum rate of change, $\lambda$ is the delay until the beginning of exponential growth, and *scal* or the scale parameter accounts for the growth (or decay) rate of the function. The logistic function is used to explain a great number of real processes related to biology, chemistry, physics, informetrics, and economy, among other fields, taking into account that it is their underlying model (*Limoges, 1987*; *Román-Román & Torres-Ruiz, 2012*). Furthermore, nonlinear mixture models are created by the addition of two or more nonlinear functions. They serve for identifying, separating, and estimating overlapping processes often accompanying further kinetic analysis (*Francisco-Fernández et al., 2012*; *Sánchez-Jiménez et al., 2013*; *López-Beceiro et al., 2010*; *López-Beceiro et al., 2012*). This is the case for the mixture logistic model:

$$m(\theta; t) = \hat{Y}(t) = \sum_{i=1}^{k} \frac{A}{1 + \exp\left(\frac{\mu_i - t}{scal_i}\right)}, \tag{10}$$

where $k$ denotes the number of logistic components, $t$ is usually time or even temperature, and $\theta$ is the vector of parameters $\theta = (A_i, \mu_i, scal_i)$. Logistic mixture models have been successfully applied in many studies on overlapping processes (e.g., degradation and crystallization processes) of different materials such as wood, polymers, composites, ceramics, and metal-organic frameworks (*Rios-Fachal et al., 2013*; *López-Beceiro et al., 2011*; *López-Beceiro et al., 2010*; *Pato-Doldán et al., 2012*; *Francisco-Fernández et al., 2012*).

## RESULTS

Once the images were obtained, four ROI classes were identified based on the information that the photographs provided about type I collagen and the extracellular matrix: type I collagen, extracellular matrix, background, and nuclei. The next step was the automatic classification of each pixel into one of the four groups or ROIs identified in the image. This approach allowed us to study the evolution of the collagen mass or extracellular matrix independently, preventing the analysis errors related to the inclusion of information corresponding to the background or artifacts (representation errors). As indicated in Section 3, the four ROIs present can be distinguished from one another according to the texture of the image. Therefore, a texture analysis was carried out in which each pixel was defined by a feature vector formed by the parameters of the structure tensor and entropy. Once each area of the images of the training sample was defined by a feature vector, a random forest supervised classification model was developed (trained) to identify the ROI corresponding to each pixel of all the images being studied. Once the results of the classifier were validated, probability maps were generated for each of the 60 images and for each above-mentioned class. Each map, estimated for each ROI, contained information on the

probability that each pixel belongs to the layer that represents the corresponding map. This section shows the main results of applying the proposed methodology to model collagen biodegradation. In accordance with the different steps of the proposed procedure, it was divided in two subsections: image analysis with feature extraction and statistical modeling.

## Image analysis and feature extraction

The probability maps corresponding to each ROI were applied as masks to the original images in order to quantify the type I collagen data extracted from each image and cellular growth and differentiation. New datasets composed of features extracted from the image (mean area in $\mu m^2$, mean circularity, the major axis of ellipses, and the mode of the greyscale histogram) were extracted from the ROIs of images corresponding to each period (measured in days).

In the case of the type I collagen group, it was observed that the probability matrices obtained by applying the random forest classifier decrease as a function of time when the ROI corresponding to type I collagen was studied, whereas those corresponding to the ROI of the extracellular matrix increase as a function of time, as expected due to osteocyte growth. These findings support the use of ROI features such as mean area for characterizing the collagen degradation due to cellular activity.

Figures 2–3 show the original micrographs and those obtained by the application of segmentation techniques. Segmentation algorithms based on detection of a threshold value and random forest machine learning algorithms were used. The segmentation of images by threshold ing, based on image conversion to the grayscale, is aimed at finding the ideal values that separate different ROIs by filtering regions of the histogram of a black-and-white image. The values of the filters can be found using local minima of the histogram (*Bulgarevich et al., 2018*). Regarding application of the random forest classifier, changes in the area and shape of objects could be observed, based on the analysis of extracellular-matrix ROIs (see Figs. 2–3). The area of the extracellular matrix seems to increase with time (violet or white area). Moreover, the region where the extracellular matrix is placed (collagen scaffold) seems to decrease because the mass of the collagen scaffold decreases with time. The extracellular matrix becomes more concentrated with time. Consequently, the degradation of type I collagen due to cellular activity and cellular growth can be studied through representative features of these images. In addition, cellular growth and the degradation degree of collagen under the influence of cellular activity are estimated by nonparametric and parametric statistical models.

To justify the use of random forest instead of other techniques such as thresholding, examples of image segmentation with two images are shown below. In Figs. 4A and 4D, two images obtained by optical microscopy are depicted, corresponding to the 18th and 38th days of observation from the beginning of the experiment, respectively. Figs. 4B and 4E present application of the threshold segmentation technique to the images corresponding to the 18th and 38th days, respectively. In addition, Figs. 4C and 4F correspond to
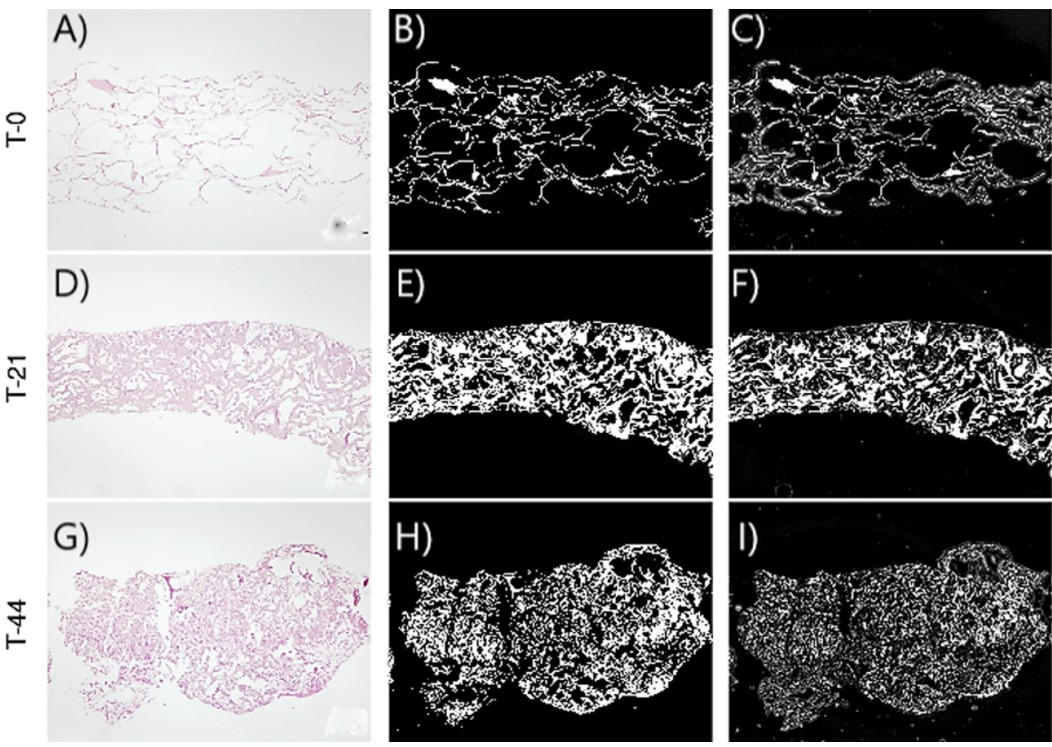

**Figure 2** **Images corresponding to cell culture samples of the CCO group: collagen, stem cells (CMMh-3A6), and the osteogenic culture medium.** The original (A, D, G), segmented images by thresholding (B, E, H), and segmented images by the random forest classifier (C, F, I) are presented. Each row refers to the time points at which the images were captured: T0 (the start of the experiment), T21 (the middle point of the experiment, 21 days), and T44 (the end of the experiment, 44 days).

segmentation of the images in Figs. 4A and 4D, respectively, by the random forest method. All the images are from the CCO group.

When the random forest and thresholding techniques were compared, we noted that there was a significant loss of data (Fig. 4E) with respect to the actual collagen shown in the image obtained from the laboratory (Fig. 4D). In fact, the image of Fig. 4D is much more difficult to segment because it contains subtle variations of intensity (color) in collagen; these variations can be detected only by the random forest technique as shown in Fig. 4F.

In addition, the segmentation technique based on random forest can be more precise, given that it allows us to obtain a collagen area that is much closer to reality. In fact, the thresholding technique also regards the nuclei and some of the extracellular matrix as a part of collagen. Therefore, the collagen area estimates by the Threshold technique are higher than what they should be.

Tenfold cross-validation techniques were applied too to measure the performance of the random forest classifier on the segmentation task in the present case study. Table 1 shows the measurements of goodness of classification of each ROI from the images in Fig. 4D. Considering only the overall precision (proportion of pixels correctly classified), 97.3% of pixels were correctly classified. This good performance is supported by Table 2, which
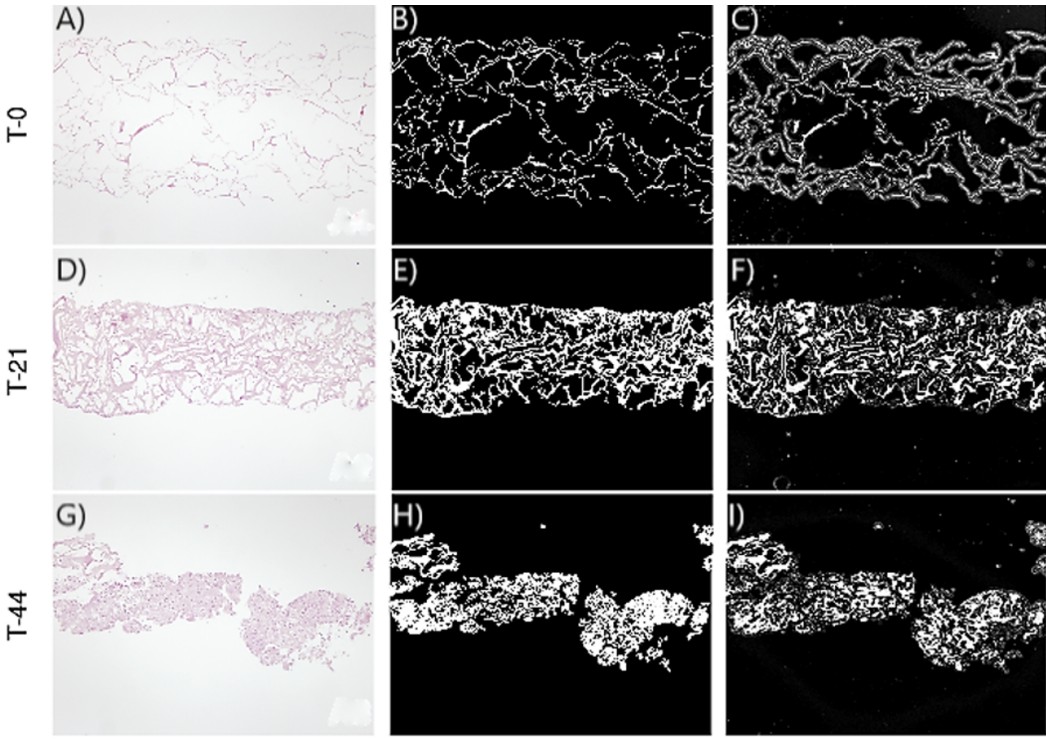

**Figure 3 Images corresponding to cell culture samples of the CCT group: collagen, stem cells (CMMh-3A6), and the nonosteogenic culture medium.** The original (A, D, G), segmented images by thresholding (B, E, H), and segmented images by the random forest classifier (C, F, I) are presented. Each row refers to the time points at which the images were captured: T0 (the start of the experiment), T21 (the middle point of the experiment, 21 days), and T44 (the end of the experiment, 44 days).

presents the confusion matrix corresponding to application of the random forest classifier to the image in Fig. 4D. The number of pixels correctly classified was much higher than that of incorrectly identified pixels for each ROI.

Figure 5 compares the threshold and random forest segmentation methods for COO and CCT groups in the range "days 0–44". The misclassification error (square root of the number of misclassified squared pixels of each image) of the threshold method tend to be greater than the corresponding to the random forest. This finding supports the use of random forest in this application.

Data, random forest parameters, segmented images and validation measures of these examples are shown in supplementary material (see Dataset S1).

Two tutorial videos are supplied in order to show the advantages of using the random forest classifier for image segmentation with respect to the threshold segmentation method. They show how to reproduce the segmentation tasks from an optical micrograph (e.g., the Fig. 4F) by random forest classifier or threshold method, using ImageJ and Weka software (see the Video S1 and Video S2).

As shown in the tutorials, the random forest classifier provides a more accurate identification of the different ROIs in an automatic way. The random forest segmentation

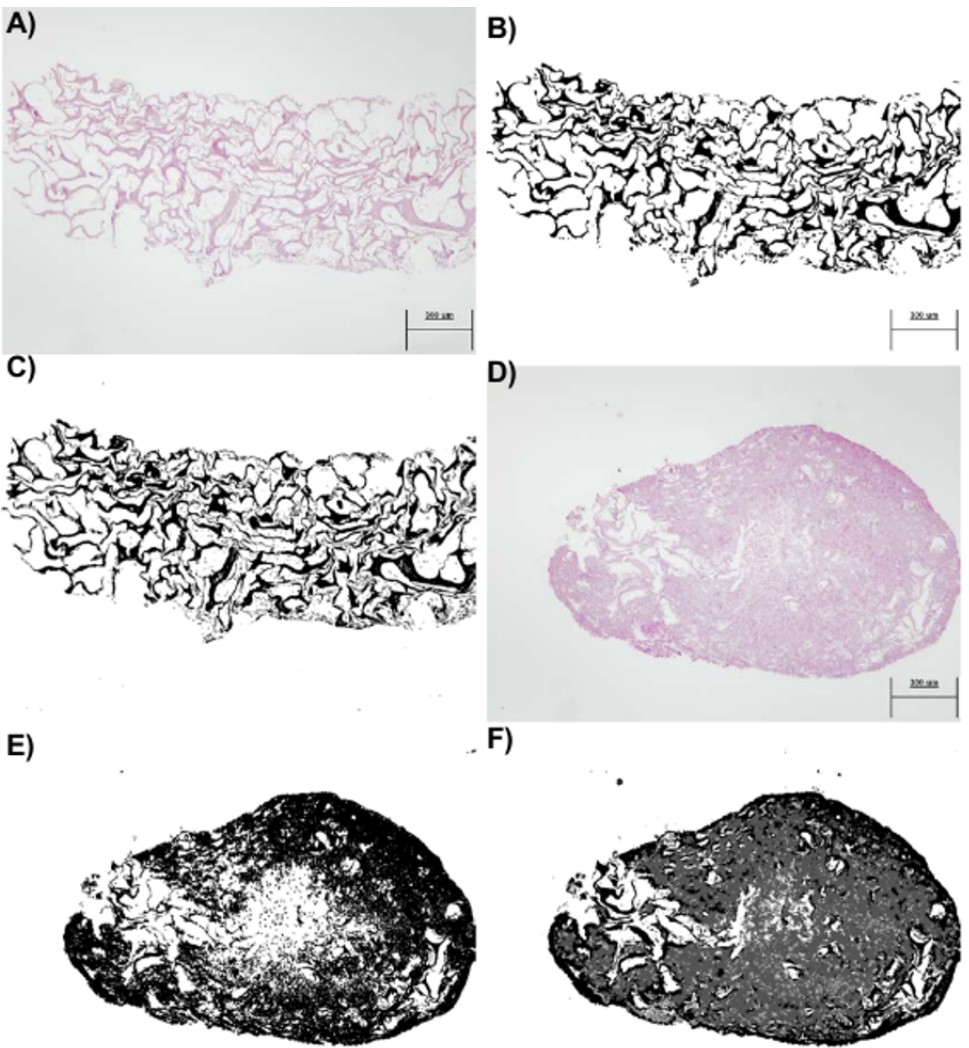

**Figure 4** (A) An image obtained by optical microscopy and belonging to the CCO group (the osteogenic medium with cells). It was obtained at time = 18 days, that is, it corresponds to an early stage, where collagen is mostly found. (B) An image of collagen resulting from the application of segmentation based on the classical threshold (Threshold) to the image of (A). The area of collagen obtained with this technique is 178965 (189 pixels represent the 300 micrometers seen in the image). (C) An image of collagen resulting from the application of segmentation based on random forest to the image of (A). The area of collagen obtained with this technique is 175261 (189 pixels represent the 300 micrometers seen in the image). (D) An image obtained by optical microscopy and belonging to the CCO group (the osteogenic medium with cells). It was obtained at time = 38 days, that is, it corresponds to a stage where a large quantity of extracellular mass and nuclei is found, in addition to a lesser quantity of collagen. (E) A collagen image resulting from the application of segmentation based on the classical threshold (Threshold) to the image of (D). The area of collagen obtained with this technique is 384170 (189 pixels represent the 300 micrometers seen in the image). (F) A collagen image resulting from the application of segmentation based on random forest to the image of (A). The area of collagen obtained with this technique is 193125 (189 pixels represent the 300 micrometers seen in the image).

**Table 1  Measurements of goodness of the ROI classification when the random forest method was applied to Fig. 4D.**

| | TP rate | Fp rate | Precision | Recall | F-Measure | MCC | ROC area | PCR area | ROI |
|---|---|---|---|---|---|---|---|---|---|
| | 0.957 | 0.008 | 0.983 | 0.957 | 0.970 | 0.957 | 0.998 | 0.997 | Collagen |
| | 0.975 | 0.023 | 0.947 | 0.975 | 0.961 | 0.944 | 0.997 | 0.993 | Extracell. matrix |
| | 0.914 | 0.004 | 0.939 | 0.914 | 0.926 | 0.922 | 0.998 | 0.979 | Nuclei |
| | 0.997 | 0.002 | 0.995 | 0.997 | 0.996 | 0.995 | 1.000 | 1.000 | Background |
| Weighted average | 0.973 | 0.010 | 0.973 | 0.973 | 0.973 | 0.963 | 0.999 | 0.996 | |

Detailed accuracy by class

**Table 2  The confusion matrix corresponding to application of the random forest classifier to the image of Fig. 4D.** The number of pixels is indicated.

Confusion matrix

| | | Predicted | | | |
|---|---|---|---|---|---|
| | | Collagen | Extracell. matrix | Nuclei | Background |
| Real | Collagen | 5727 | 220 | 25 | 10 |
| | Extracell. matrix | 92 | 5630 | 36 | 16 |
| | Nuclei | 7 | 90 | 1073 | 4 |
| | Background | 1 | 7 | 9 | 6361 |

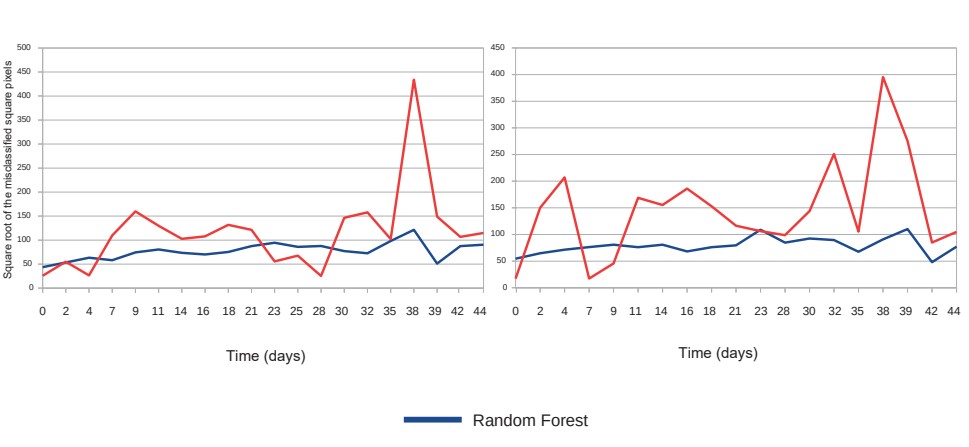

**Figure 5  (A) Square root of the number of misclassified squared pixels for each image (when collagen ROI is analyzed) obtained at each time point and corresponding to the CCO group; (B) square root of the number of misclassified squared pixels for each image (when collagen ROI is analyzed) obtained at each time point and corresponding to the CCT group.**

method is more useful to identify the different regions of interest, even when the differences between regions are very slight and can be overlapped with the noise of the image.

Conversely, the threshold method has some drawbacks:

1. It is necessary to manually adjust the thresholds corresponding to the different ROIs, and

2. this can lead in the losing of important information related to the area and edges of the different ROIs;

3. in addition, threshold method is highly depended on the operator skills (not as automatic as the random forest method).

Summarizing, out of two techniques, the random forest classifier is the more automatic and accurate alternative for segmentation task, preventing in a greater extent undesirable effects such as the image noise.

## Statistical regression modeling

The first step is to apply descriptive analysis techniques in order to evaluate the dependence structure between the variables critical for collagen degradation and osteocyte growth.

Then, the collagen degradation modeling task was done. Further regression analysis of image feature variables related to cell growth and differentiation is necessary to determine the influence of cell growth on collagen degradation.

Finally, the cell growth influence on collagen degradation was modeled via the degree of collagen degradation (measured by the mass loss percentage) and the mean area, mean circularity, and grayscale histogram mode corresponding to each image captured at each time point.

### *Exploratory correlation analysis*

Identifying and characterizing dependence relations between features extracted from an image, time, and the degree of degradation (collagen mass loss) are the goals here.

Scatterplots are the most intuitive way to check the dependence structure of a dataset. Figure 6 illustrates the scatterplot matrix for mean area, mean circularity, collagen mass loss, and time for groups CCO and CCT. Other interesting variables (such as the major axis of the fitted ellipse or even mean roundness of the objects in each image) were not included due to their strong dependence on the area or circularity. The area and circularity of the extracellular matrix provide information about cellular growth and shape.

The strongest dependence was found between the mass loss and time variables. In fact, the collagen mass loss strongly depends on time in a relatively complex nonlinear way. It seems that collagen degradation is composed of two main steps that could be related to two main degradation processes. This trend was observed in both groups CCO and CCT, with only slight differences between them.

In addition, an asymptotic type of dependence between circularity and time was observed, whereas mean area seemed to increase with time. This increase seemed to be of the sigmoid type when the CCO group was studied, but in the case of the CCT group, the type of trend was not clear due to high dispersion.

If the dependence structure of the features of the CO group is studied and compared with that of groups CCO and CCT (see Fig. 7A), we note that there are no trends between

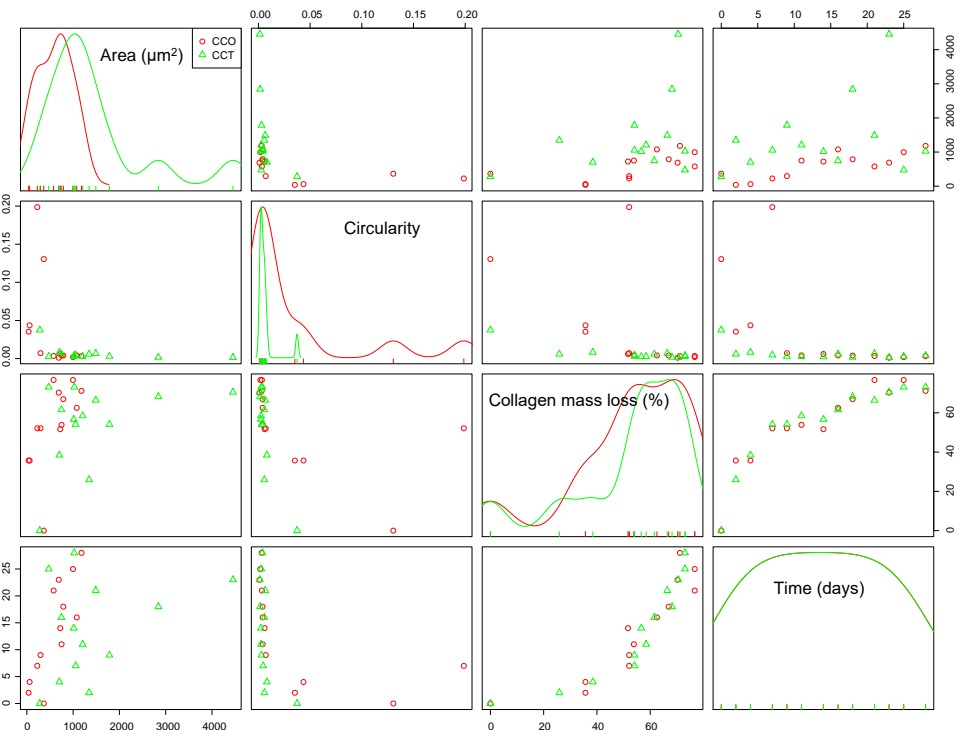

**Figure 6** The scatterplot matrix corresponding to the mean area, mean circularity, collagen mass loss, and time within the CCO and CCT groups.

variables (either linear or nonlinear), and scattering is substantial. This finding is in agreement with the fact that CO is the control group without stem cells.

Furthermore, Fig. 7B depicts the relations between the variables when a logarithmic transform is applied to the mean area and mean circularity. We see that there could be linear relations between the logarithmic transform of the area and circularity as a function of time. In addition, there could be slightly linear relations of the collagen mass loss with the logarithmic transform of the area and circularity. This information will help to estimate the proper statistical models for the degree of collagen degradation.

### Collagen degradation modeling

Estimating the underlying model that explains the degree of collagen degradation depending on time is the goal here. This model can provide information about the different steps of degradation and can serve for forecasting tasks. The mass loss feature is the critical variable chosen for characterizing the degree of type I collagen degradation.

According to the results of the exploratory analysis, the use of a sigmoid-type nonlinear function is needed to describe the relation between mass loss and time. Moreover, it seems that the main trend can be the result of the sum of two sigmoid trends.

A nonparametric GAM with penalized regression splines was applied to confirm the results of descriptive analysis. In fact, in Figs. 8A and 8B, the GAM estimates for groups CCO and CCT are shown. Taking into account the fit and bootstrap confidence intervals, we infer

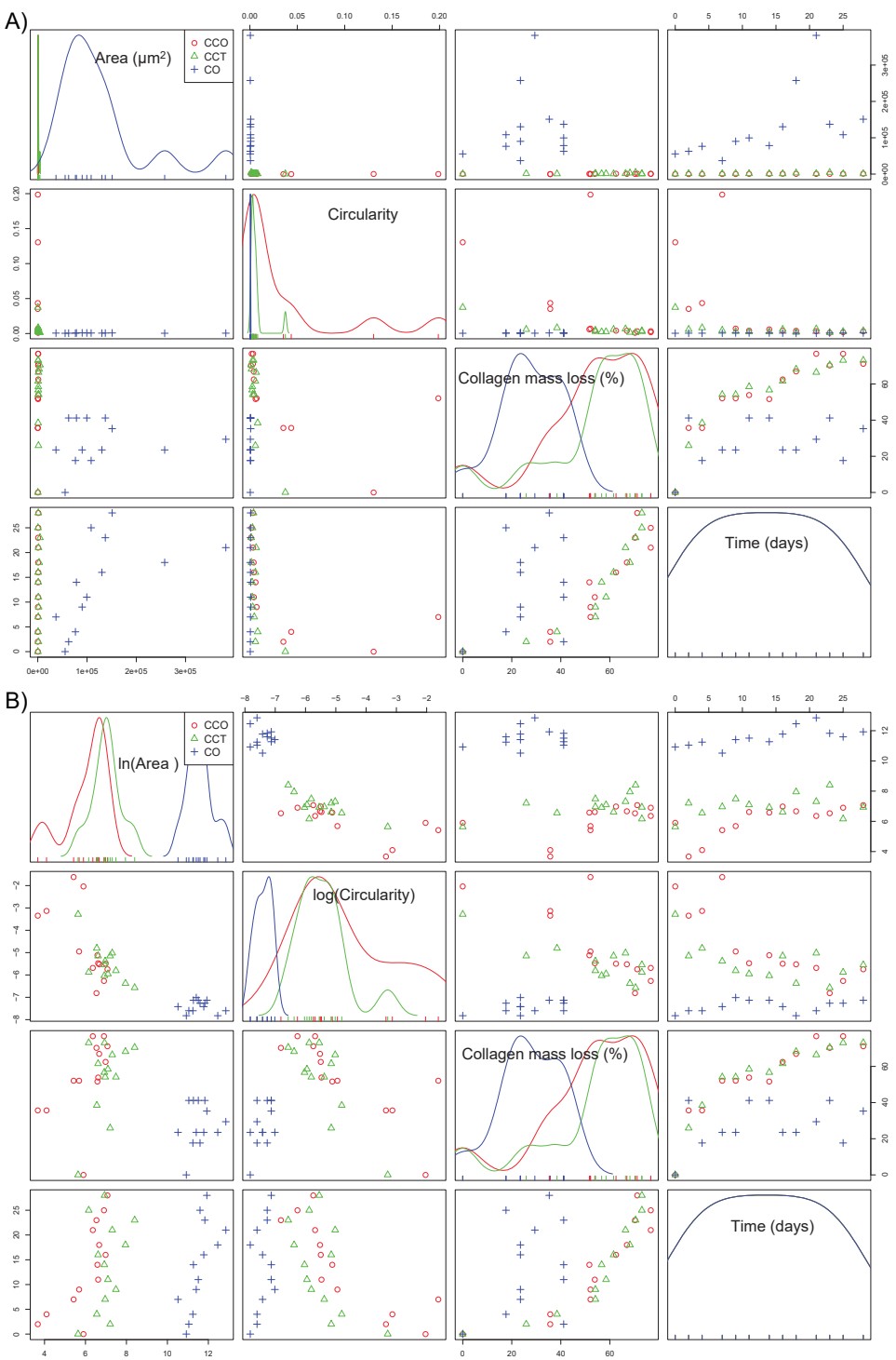

**Figure 7** (A) The scatterplot matrix corresponding to the mean area, mean circularity, collagen mass loss, and time within groups CO, CCO, and CCT; (B) the scatterplot matrix corresponding to the neperian logarithm of the mean area, neperian logarithm of mean circularity, collagen mass loss, and time within groups CO, CCO, and CCT.

that the main trend could be composed of two sigmoid curves, each one corresponding to different degradation processes.

A more accurate fit was obtained for the CCT group and explained 99% of the collagen mass loss information. In any case, it seems that we can obtain accurate estimates of mass loss as a function of time taking into account the high values of the determination coefficient and the relatively narrow confidence intervals.

The next step was to fit a parametric model based on the adequate nonlinear function. The aim was to obtain a transfer function that, in addition to interpretable parameters, could provide predictions and allow us to compare different groups. The sigmoid trend and the fact that the mass loss increases monotonically support the use of logistic functions. In accordance with the GAM results, a nonlinear regression model based on a mixture of two logistic functions (defined by three parameters) is proposed. This type of nonlinear regression model has been successfully applied to separate overlapping degradation processes within the framework of thermal analysis in degradation studies (*Tarrío-Saavedra et al., 2014*). Figures 8C and 8D shows the logistic-mixture-model fittings that estimate the mass loss depending on time for groups CCO and CCT, respectively. Estimation and prediction asymptotic confidence intervals are provided too. Table 3 shows signification analysis and the model parameters. Signification analysis, fitted trends, confidence intervals, and the achieved goodness of fit ($R^2 = 0.94$ and $R^2 = 0.99$ for groups CCO and CCT, respectively) justify the use of the logistic mixed model to explain the mass loss of collagen. The first component is characterized by an inflection point at $\sim$2 days and a saturation asymptote at 50% of collagen mass loss. On the contrary, the second logistic component is defined at an inflection point of $\sim$16–17 days and an overall mass loss of between the 21–13 of weight percent loss. The meaning of the two logistic components can be guessed taking into consideration control sample CO. In fact, the samples of the CO group lost mass almost immediately, and the mass remained constant at $\sim$30% mass loss. Thus, this mass loss is related to another factor apart from cellular activity. Deeper research is necessary to identify the nature of the degradation process (either hydrolytic or otherwise). In addition, we infer that the first logistic component depends, to a large extent, on another degradation factor aside from cellular differentiation and growth. In fact, considering the porosity of collagen, the degradation due to cell growth and differentiation should take place mainly after longer periods. Regarding the second logistic component, it may be more related to cell activity (given that the inflection point is $\sim$16–17 days) that corresponds to $\sim$ 20% collagen mass loss after 28 days.

The progression of degradation of type I collagen can be compared between groups CCO and CCT through examination of the logistic components. Figure 8E shows the fitted first logistic component for groups CCO and CCT. There are only slight differences, but it seems that the degradation process tends to begin even before the inflection point $\mu_{CCO}(= 1.73\ days) < \mu_{CCT}(= 2.47\ days)$. The results on bootstrap confidence intervals for parameter $\mu$ (implemented using the `nlstools` package (*Baty et al., 2015*) with 1,000 resamplings) suggest that CCO samples tend to begin to degrade before the CCT samples do. There are, however, no differences between the groups in terms of the mass loss percentage and the rate of mass loss for this first degradation process. Furthermore, in

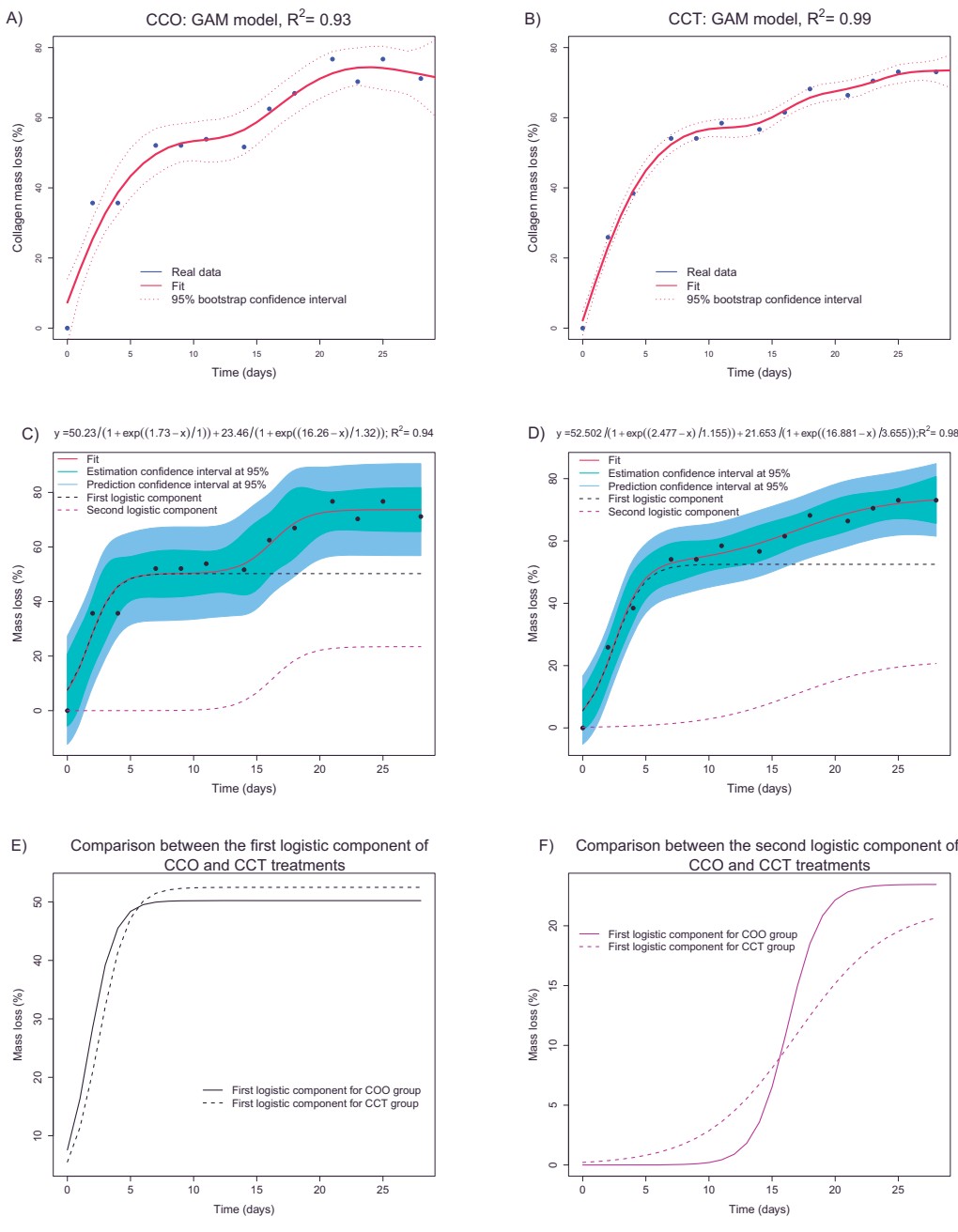

**Figure 8** (A and B) depict the fittings (via a nonparametric GAM) of mass loss as a function of time in groups CCO and CCT, respectively. (C and D) show the fittings of the parametric logistic mixture model to estimate the mass loss with respect to time in groups CCO and CCT, respectively. (E and F) illustrate the two logistic components of the logistic mixture model applied to groups CCO and CCT, respectively.

**Table 3** The signification analysis of fitted parameters of the mixed logistic model and bootstrap intervals corresponding to groups CCO and CCT at the 95% confidence level.

| Parameters | Estimates | Bootstrap 95% interval lower limit | Bootstrap 95% interval upper limit | Standard error | t value | p value |
|---|---|---|---|---|---|---|
| **CCO** | | | | | | |
| $A_1$ | 50.223 | 42.841 | 54.690 | 3.900 | 12.88 | 4e−06 |
| $\mu_1$ | 1.7340 | 1.0042 | 2.4513 | 0.481 | 3.604 | 0.0087 |
| $scal_1$ | 1.0000 | 0.4259 | 1.6945 | 0.446 | 2.240 | 0.0600 |
| $A_2$ | 23.463 | 17.667 | 34.728 | 5.463 | 4.295 | 0.0036 |
| $\mu_2$ | 16.261 | 14.273 | 18.803 | 1.329 | 12.24 | 5.6e−6 |
| $scal_2$ | 1.3233 | 0.3400 | 4.1071 | 1.245 | 1.063 | 0.3230 |
| **CCT** | | | | | | |
| $A_1$ | 52.502 | 36.131 | 58.400 | 7.699 | 6.819 | 0.0003 |
| $\mu_1$ | 2.4768 | 2.0578 | 3.0790 | 0.370 | 6.694 | 0.0003 |
| $scal_1$ | 1.1546 | 0.5631 | 1.6311 | 0.337 | 3.430 | 0.0110 |
| $A_2$ | 21.653 | 13.539 | 55.246 | 12.52 | 1.729 | 0.1274 |
| $\mu_2$ | 16.881 | 10.753 | 26.856 | 3.040 | 5.553 | 0.0009 |
| $scal_2$ | 3.6554 | 0.6866 | 10.410 | 3.777 | 0.968 | 0.3654 |

Fig. 8F, the second logistic components of groups CCO and CCT are compared. The rate of degradation corresponding to the CCO group is significantly greater than that corresponding to the CCT group. This finding can be interpreted via analysis of the *scale* parameter. In fact, the bootstrap confidence interval for the scale parameter is placed at lower values than those corresponding to group CCT (at lower values of *scale*, the logistic curve has a higher rate of change). Therefore, the osteogenic medium slightly promotes the degradation of collagen, advancing the degradation process and increasing the rate of degradation.

### Cell growth modeling

The mean area and mean circularity of the identified objects of the extracellular matrix obtained from each image at each time point provide important information about cell growth and differentiation. Accordingly, the dependence relation of the area and circularity with time was characterized and statistically modeled. Figure 9 shows the best models fitting the area and circularity variation depending on time, which were chosen according to goodness of fit and parameter signification criteria.

### Collagen degradation depending on cell growth modeling

The next step is to determine the dependence relation between collagen degradation (through its mass loss percentage) and cell growth and cell differentiation indicators such as mean area and circularity of the extracellular matrix.

On the basis of the results in the previous sections, logarithmic transformation of the area and circularity was proposed above to obtain more informative, simpler, and more accurate models. Indeed, linear regression models are proposed to explain the collagen mass loss with respect to circularity and the area (after natural logarithms are applied). Figure 10 presents different linear models fitted to the real data obtained from groups CCO

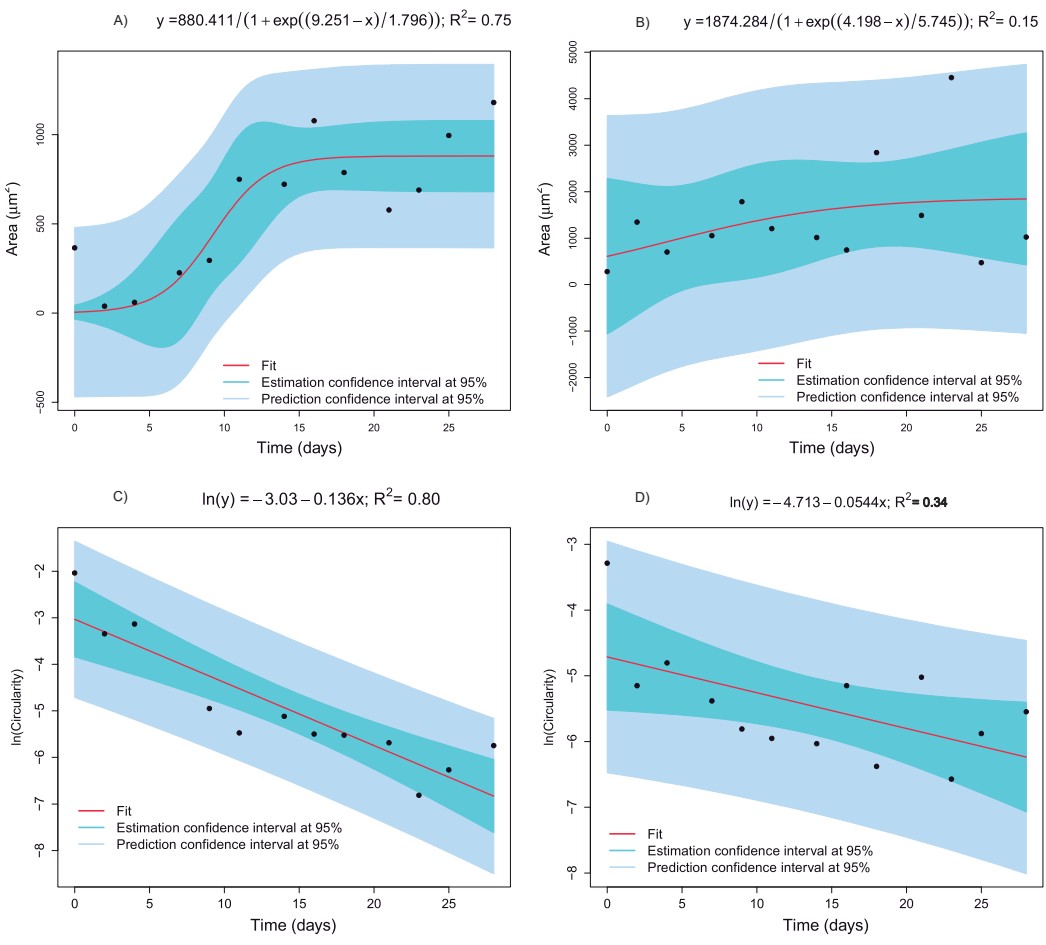

**Figure 9** **Regression models fitted to the mean area or mean circularity of extracellular-matrix images as a function of time.** (A) Mean area as a function of time corresponding to the CCO group (logistic model). (B) Mean area as a function of time in the CCT group (logistic model). (C) The natural logarithm of mean circularity of the extracellular matrix as a function of time in the CCO group (linear model). (D) The natural logarithm of mean circularity of the extracellular matrix as a function of time in the CCT group (linear model).

(Figs. 10A, 10C) and CCT (Figs. 10B, 10D). Very informative linear models were fitted to estimate the mass loss versus area ($mass\ loss = -5.217 + 10.578\ln(area)$ with $R^2 = 0.70$) and depending on circularity ($mass\ loss = -19.4 - 14.9\ln(circularity)$ with $R^2 = 0.88$) for the CCO group. It is important to stress the strong relation between the mass loss and circularity. The obtained model could be useful even for prediction tasks. The same trends were found in the CCT group, but high dispersion prevented obtaining more informative models. In conclusion, the degree of degradation, measured as the mass loss percent, is proportional to the logarithm of extracellular-matrix area and inversely proportional to the logarithm of cell circularity.

A more complete multivariate linear model that accounts for all the collagen degradation sources of variation was also estimated and validated by a threefold cross-validation

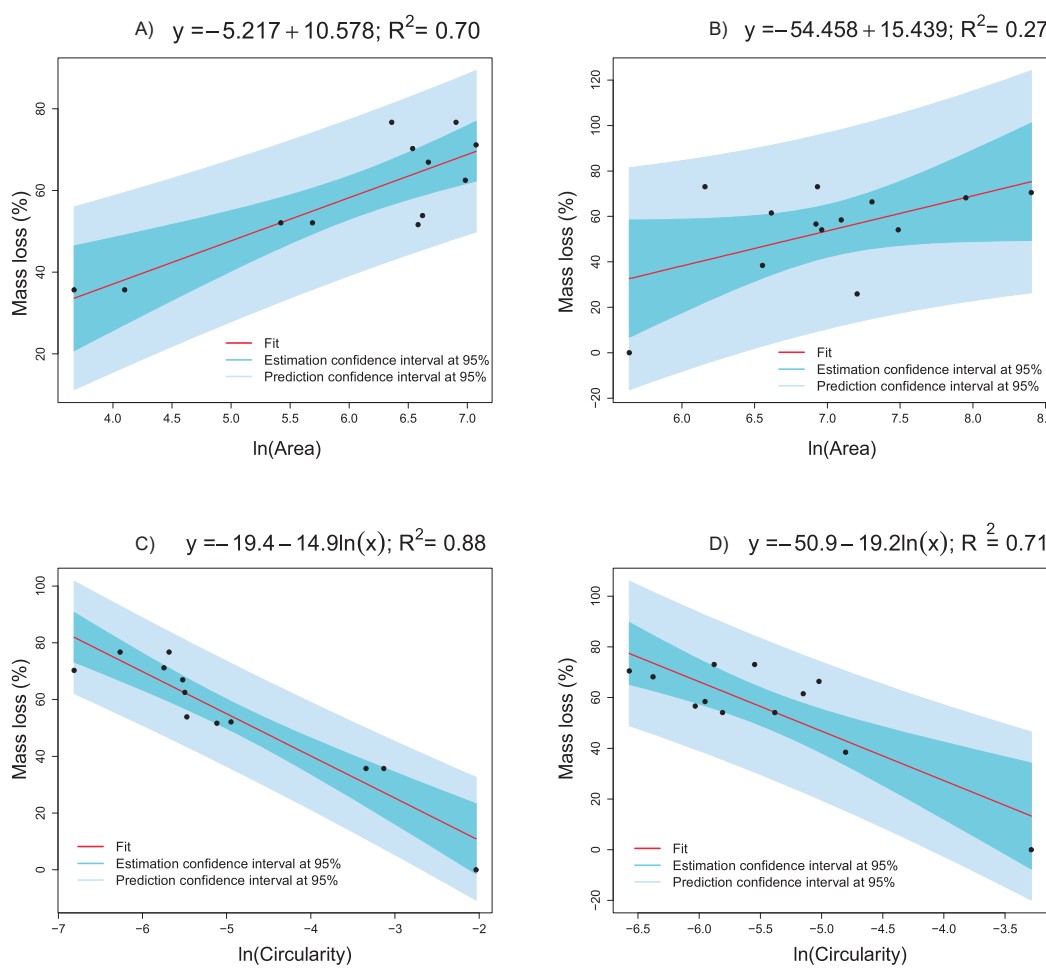

**Figure 10** **Linear models fitted to the collagen mass loss obtained in groups CCO and CCT.** (A) Collagen mass loss as a function of the natural logarithm of mean area of the extracellular matrix in the CCO group. (B) Collagen mass loss as a function of the natural logarithm of mean area of the extracellular matrix in the CCT group. (C) Collagen mass loss as a function of the natural logarithm of mean circularity of the extracellular matrix in the CCO group. (D) Collagen mass loss as a function of the natural logarithm of mean circularity of the extracellular matrix in the CCT group.

procedure. The estimated model expression (see Table 4) is

$$\widehat{Mass\ loss} = -17.0396 - 0.215CCT - 2.1751CO - 0.0599\ln(circularity)$$
$$+ 3.7804\ln(mode) + 0.1760\ln(area)$$

where $\widehat{Mass\ loss}$ is the mass loss estimates; $CCT$ and $CO$ are dichotomous variables equal to 1 if the corresponding group is CCT or CO, respectively; and *mode* is the grayscale histogram mode corresponding to each image. It is characterized by $R^2 = 0.89$. The latter variable is included to take into account information about the image color (possibly related to cellular growth) in the model. Increasing the extracellular-matrix area and histogram mode increases the collagen mass loss, whereas decreasing circularity increases the mass

**Table 4  Signification analysis of multivariate-linear-model parameters and bootstrap intervals at the 95% confidence level.**

| Parameters | Estimates | Bootstrap 95% interval lower limit | Bootstrap 95% interval upper limit | Standard error | $t$ value | $p$ value |
|---|---|---|---|---|---|---|
| Intercept | −17.0396 | −25.88 | 2.79 | 4.5 | −6.1 | 2.3e−6 |
| CCT | −0.2150 | −0.346 | −0.078 | 0.07 | −3.2 | 0.0033 |
| CO | −2.1751 | −2.587 | −1.641 | 0.23 | −9.6 | 7e−10 |
| ln(circularity) | −0.0599 | −0.173 | 0.008 | 0.0514 | −1.7 | 0.1080 |
| ln(mode) | 3.7804 | 2.509 | 5.387 | 0.52 | 7.2 | 1.5e−7 |
| ln(area) | 0.1760 | 0.065 | 0.249 | 0.04 | 4.1 | 0.0004 |

loss. The effects in CO and CCT groups are a decrease in the collagen mass loss due to cell activity, when compared with the CCO reference group.

Figure 11A illustrates relative importance of the group, extracellular-matrix histogram mode, circularity, and area for the type I collagen mass loss, in % of $R^2$. For this purpose, the $R^2$ contribution averaged over orderings of regressors and lmg metrics (*Grömping, 2006*) were used. Almost all the variability in the collagen mass loss could be explained by group (∼56% of $R^2$) and mean area variation (∼23% of $R^2$) (with preapplied logarithm transformation), but the contributions of circularity (∼8% of $R^2$) and mode (∼14% of $R^2$) are significant too. The overall determination coefficient of the multivariate linear model is 0.89.

Figure 11B shows the results of the cross-validation procedure for measuring predictive accuracy of the multiple linear regression model. The data are randomly assigned to three different folds. During three iterations, each fold is removed (test sample), while the remaining data (training sample) are employed to evaluate the model and then to predict the mass loss corresponding to the removed observations (corresponding to the test group). In fact, observed versus predicted values are depicted in a scatterplot. The cross-validation predictions for each of the three groups in which the data are divided are indicated with different colors. The data points revealed linear trends almost coincident with each other and very close to the bisector. These findings support high predictive accuracy of the newly developed multivariate model for analysis of collagen mass loss. In conclusion, prediction of the degradation degree of collagen (measured as the percentage of mass loss) could be done via the predictors based on image analysis of the extracellular matrix.

Data, workplace and scripts corresponding to the statistical analysis can be found in Dataset S2, Dataset S3 and Code S1.

## DISCUSSION

The combination of microscopy, image analysis, classification machine learning methods, and statistical learning tools of regression modeling allowed us to model the degradation level of collagen as a function of time and depending on cell growth and differentiation features. Consequently, the main goal of this work is to provide a systematic and quantitative procedure for estimating the degree of degradation of collagen scaffolds. This work includes both (1) a proposed strategy for obtaining relevant data by imaging techniques, and (2) a

**A) Relative importances of predictors for the logarithm of collagen mass loss**

**with 95% bootstrap confidence intervals**

**Method LMG**

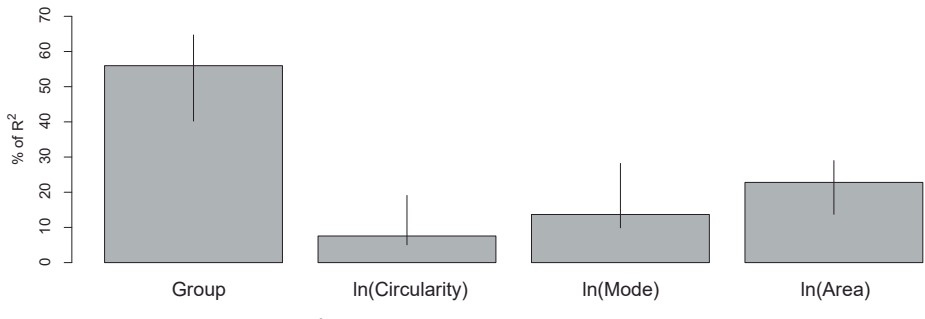

R²=88.7%, metrics are normalized to sum 100%

**B) Cross-validation measures of predictive accuracy for multi e linear regression**

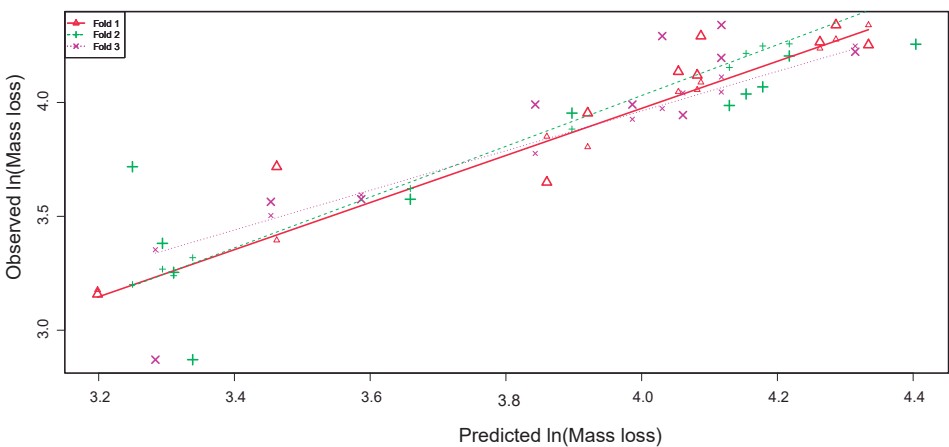

**Figure 11** (A) Relative importance of predictors (in terms of $R^2$; in the multivariate linear model) that explain the collagen mass loss; (B) observed values vs. model predictions when 2/3 of data was utilized as a training sample via a cross-validation procedure to evaluate predictive accuracy of the model.

comprehensive statistical approach to model the collagen degradation and cell growth. This approach provides information complementary to the results of relevant works where the collagen degree of degradation has been studied but not modeled (*Alberti & Xu, 2015*; *Ma et al., 2003*). The statistical modeling also provides tools for characterizing degradation of a material and for separating the effects of different sources of degradation (hydrolysis and biological degradation). In addition, the relation between mass loss and features extracted from micrographs (using optical microscopy) was studied here, opening up a way to characterize collagen degradation in a more automatic manner by image analysis. Besides, this methodology could be useful for research into the degradation of scaffolds other than type I collagen (*Alizadeh et al., 2013*; *Han et al., 2014*).

First of all, the degree of degradation of type I collagen was studied, characterized, and modeled via experimental data (see Fig. 8). The mass loss percentage was found to be a critical variable for collagen degradation, and its dependence on time was studied. The statistical exploratory analysis and application of a GAM revealed that there are at least two sigmoid processes of degradation. Thus, a logistic mixture regression model was proposed and successfully applied to determine this relation, which represents a mode of degradation of collagen. The first logistic component (it accounts for 50% of collagen mass) depends on another degradation factor apart from cellular differentiation and growth. Further research is needed to identify the type of the degradation processes involved, namely a hydrolysis process (*Schliecker et al., 2003*). The second logistic component may be more clearly related to the cell activity and corresponds to 20% of collagen mass. Differences between fittings corresponding to groups CCO and CCT were uncovered by studying the first and second logistic functions. Thus, the osteogenic medium plays a significant role in collagen degradation. Namely, the first logistic degradation process tends to take place earlier in the CCO group than in group CCT. In addition, the rate of degradation in the second logistic degradation process is higher in the CCO group. This comparison was made via examination of model parameter estimates such as logistic inflection point $\mu$ and *scale* (see Table 3). The logistic mixture models could help to separate the degradation modes due to different causes (cell growth and hydrolysis among others) and to study the collagen biodegradation more properly.

Once the degree of degradation of collagen was modeled as a function of time, this study also addressed the problem of estimating the progression of collagen degradation with respect to the growth and differentiation of osteocytes from stem cells. This relation modeling is necessary to develop tissue engineering applications. Therefore, methodologies that can extract relevant quantitative information about cell growth are necessary. The application of microscopy, combined with image analysis and supervised classification, is proposed in this work to perform this task. In fact, the random forest supervised classification algorithm permits the identification of different regions of interest in each micrograph automatically: extracellular matrix, collagen, nuclei, and background. Thus, the extracellular matrix areas of each image and their evolution depending on time can be studied separately. The random forest methodology can be applied if previous image texture analysis and feature extraction (pixel energy and tensor matrix) can be implemented. From segmented images of the extracellular matrix, another feature extraction process is implemented to obtain characteristics such as the mean area and mean circularity of the identified object in each image. These are relevant features of osteocyte cell growth. Images show how the extracellular matrix ROI increases in terms of area, whereas the collagen region (that supports the cells) decreases. This can be observed in Figs. 2–4–9, in which the region containing the extracellular matrix decreases, although its concentration (and its area) increase significantly. This relevant information about cell growth is summarized in the extracted features shown in Figs. 6 and 7.

The statistical modeling of representative features of cell growth and differentiation with respect to time is intended. Identifying the type of relationship allow us to obtain information about the degree of degradation due to cell activity. In the case of the CCO

group, the area varies in a sigmoid way with respect to time, in fact the logistic model explain the 75% of the overall variability (Fig. 9A). The bootstrap 95% confidence interval for $\mu$ parameter, $6.756 < \mu < 12.177$, shows that the extracellular mean area growth could be more related with the second logistic function of the fitting of mass loss versus time. Thus, this second logistic component is also related with the biological degradation. Moreover, the evolution of the image objects mean circularity is studied. Taking into account its asymptotic decay, a logarithmic transformation is proposed. A clear linear trend can be observed in the Fig. 9C, indeed the fitted linear model $\ln(circularity) = -3.03 - 0.136\ Time$ explain the 80% of $\ln(circularity)$. Therefore, the extracellular matrix circularity is time dependent. The same trends can be observed for area and circularity of extracellular matrix of CCT group but with higher dispersion. In fact, the measurements uncertainty prevents obtaining an informative model for the area depending on time.

Informative linear regression models were fitted to define the relationship between collagen degradation and cellular growth (Fig. 10). The collagen mass loss linearly varies with respect to area and circularity when logarithmic transformation of predictors is applied. In fact, mass loss increases significantly at a constant rate when $\ln(area)$ increases. Otherwise, mass loss decreases significantly at a constant rate when $\ln(circularity)$ increases. Thus, increasing osteocyte extracellular matrix area and decreasing its circularity are related with the increase in collagen mass loss, i.e., the degradation degree. When the circularity of the extracellular matrix is known, the scaffold mass loss (and thus the degree of degradation) can be estimated through the application of linear models, mainly when the CCO group is studied ($R^2 = 0.88$). A relatively strong linear dependence was also identified between mass loss and extracellular matrix area ($R^2 = 0.7$), providing a tool to estimate the degree of collagen degradation in a reasonable way. In the CCT group, the linear relationships between mass loss with respect to extracellular matrix area and circularity are weaker than in the CCO group (Fig. 10). Thus, the osteogenic medium is an influencing factor of the degree of degradation.

Furthermore, a multivariate linear regression model was proposed to estimate the collagen mass loss for all the groups studied (including or not osteogenic medium and/or stem cells), including also the area, circularity and grayscale histogram mode predictors (see Table 4). This model allows us to estimate the degree of degradation taking into account a comprehensive set of sources of variation. The fitted model explains about the 89 % of collagen mass loss total variability. The more influencing predictors on collagen degradation are the group and area. This means that the collagen degradation mainly depends on the variation of osteogenic medium and the addition of stem cells. Moreover, the area of extracellular matrix accounts for more than the 20% of the explained model variance. Thus, the collagen degradation (mass loss) is highly dependent on cell growth. In a less extent, histogram mode and circularity also influence the level of degradation of collagen. Finally, the mass loss prediction accuracy of the model was successfully tested by a 3-fold cross validation process (Fig. 10). Thus, this model could be used to estimate the collagen degree of degradation from the features obtained from image analysis. In conclusion, the prediction of the degree of degradation of collagen (measured as the
percentage of mass loss) could be done from the predictors based on the image analysis of extracellular matrix.

## CONCLUSIONS

A methodology based on image analysis, random forest classification, and statistical learning was proposed for estimating the degree of degradation of collagen scaffolds due to osteocyte cell growth and differentiation. The mass loss percentage is defined as the critical variable for collagen degradation and its dependence on time and features extracted from micrographs (extracellular matrix area, circularity, histogram mode) was studied.

The random forest classifier has been proposed to perform the segmentation task, i.e., the identification of the different ROIs. In fact, the random forest classifier is a more automatic and accurate alternative for identifying the different ROIs (collagen, extracellular matrix, nuclei) than the classical thresholding method, preventing in a greater extent undesirable effects such as the image noise.

The statistical descriptive analysis and the nonparametric GAM regression provide information about two different degradation processes. They are identified as two overlapped sigmoid type steps in the type I collagen mass loss trend. Consequently, a logistic mixture regression model was proposed to define this relationship that represents the path of degradation of collagen. Using this type of statistical models allows the separation of the effects of different types of degradation. The first logistic component accounts for other degradation procedures apart from cellular differentiation and growth, whereas the second logistic component is more related to the cell activity, which corresponds with around the 20% collagen mass. Moreover, by studying the parameters of the first and second logistic components of the model, differences between CCT and CCT groups were found. Accordingly, the osteogenic medium was found to be an influencing factor in collagen degradation and its influence could be estimated quantitatively.

The relationship between the degree of collagen degradation with respect to the growth and differentiation of osteocytes from stem cells was also modeled. The combined use of microscopy, image analysis, and random forest supervised classification was proposed to extract relevant quantitative information about cell growth. The different regions of interest of each micrograph were identified automatically: extracellular matrix, collagen, and background artifacts. The extracellular matrix areas (representative of cell growth) of each image and their evolution depending on time can be studied separately. As a result, the area, circularity and histogram mode, among other relevant features, were extracted to summarize the osteocyte growth information. The evolution of these features with respect to time was modeled using nonlinear (based on logistic function), and linear models (after logarithmic transformation of predictor).

The relation between the advancement of collagen degradation and cellular growth was modeled. Increases in osteocyte extracellular matrix area and decreases in its circularity are related to the increase in the collagen mass loss and, thus, the degree of degradation. The mass loss changes linearly way with respect to the extracelular matrix area and circularity when logarithmic transformation is applied. These relationships were modeled through

linear regression models. The mass loss versus circularity relationship was found to be stronger than those between mass loss and extracellular matrix area. This could be because the increase in the extracellular matrix area was related to collagen degradation only when collagen pores are filled.

A multivariate linear regression model was used to estimate more accurately the degree of collagen degradation, including all the influencing variables, in addition to evaluating the importance of each predictors. The osteogenic medium type (and stem cell addition) and extracellular matrix area are predictors exerting the greatest influence on collagen degradation. Taking into account that the area accounts for more than the 20% of the $R^2$, collagen degradation is highly dependent of cell growth. After prediction accuracy testing, this model could be used to estimate the collagen degree of degradation from the features obtained from image analysis.

### Funding

This work has been supported by MINECO grants MTM2014-52876-R and MTM2017-82724-R, and by the Xunta de Galicia (Grupos de Referencia Competitiva ED431C-2016-015 and Centro Singular de Investigación de Galicia ED431G/01), all of them through the ERDF. The research of Yaroslava Robles has been supported by the Ecuador's Secretaría Nacional de Educación Superior, Ciencia, Tecnología e Innovación (SENESCYT) and Inditex-UDC International Doctoral School Grant for pre-doctoral students. The funders had no role in study design, data collection and analysis, decision to publish, or preparation of the manuscript.

### Grant Disclosures

The following grant information was disclosed by the authors:
MINECO: MTM2014-52876-R, MTM2017-82724-R.
Xunta de Galicia (Grupos de Referencia Competitiva ED431C-2016-015 and Centro Singular de Investigación de Galicia ED431G/01).
Ecuador's Secretaría Nacional de Educación Superior, Ciencia, Tecnología e Innovación (SENESCYT).
Inditex-UDC International Doctoral School Grant for pre-doctoral students.

### Competing Interests

The authors declare there are no competing interests.

### Author Contributions

- Yaroslava Robles-Bykbaev conceived and designed the experiments, performed the experiments, analyzed the data, prepared figures and/or tables, authored or reviewed drafts of the paper.
- Salvador Naya conceived and designed the experiments, analyzed the data, authored or reviewed drafts of the paper, approved the final draft.

- Silvia Díaz-Prado conceived and designed the experiments, performed the experiments, contributed reagents/materials/analysis tools, approved the final draft.
- Daniel Calle-López analyzed the data, prepared figures and/or tables.
- Vladimir Robles-Bykbaev analyzed the data, prepared figures and/or tables, authored or reviewed drafts of the paper.
- Luis Garzón analyzed the data, contributed reagents/materials/analysis tools.
- Clara Sanjurjo-Rodríguez conceived and designed the experiments, performed the experiments, contributed reagents/materials/analysis tools.
- Javier Tarrío-Saavedra conceived and designed the experiments, analyzed the data, prepared figures and/or tables, authored or reviewed drafts of the paper, approved the final draft.

## Data Availability

The raw measurements are available in the Supplemental Files.

## Supplemental Information

Supplemental information for this article can be found online at http://dx.doi.org/10.7717/peerj.7233#supplemental-information.

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
