# Peer review of "An artificial-vision- and statistical-learning-based method for studying the biodegradation of type I collagen scaffolds in bone regeneration systems"

_PeerJ, doi:10.7717/peerj.7233_

## Round 0.1 · original submission · Major Revisions

Both reviewers find the manuscript interesting and raise concern regarding to the results validation. In addition, writing should be improved.

Reviewer 1 ·

Basic reporting

The English is ambiguous, the introduction statement needs references for justification.

The English is ambiguous, for example, line 33-38, on the other hand, ..., on the other hand line 94: The computational tool used ... has been the statistical software R (R Core Team, 2017). line 108-109, line 141-144, 254 etc.

In sufficient literature reference, line 51-64, general statements without reference.
line 68, image segmentation methods are listed, but no further introduction, pros/cons.

line 91, the R squared definition equation in a paragraph, do we really need this?

line 108-112 general statements no reference.

line 125-126: the above ... procedures, even though all are valid, are not usually applicable for all cases. This is not a professional description.

line 407: the linear models are the most simple and usual parametric models, but in chemistry and biology domains, the nonlinear models are also very important and useful. Any reference, example to justify this general statement?

Equation numbers are missing: line 390, 394, 401, 412, etc.

Figure 7, label small letter inconsistent with the capital letters labelled in the figure.

Experimental design

The overall experimental design is proper, from image preprocessing, to feature analysis, such as correlation and time dependency modeling/analysis among features, and the regression model fitting for collagen degradation.

The procedure to obtain the image data is clear, the paper needs more introduction and justification on using the RandomForest, the pros and cons in comparison with other machine learning methods.

The paper modeled and examined time dependency of collagen mass loss, mean area and mean circularity in a systematical way. Some interesting patten was found, such as the two sigma models for collagen mass loss (Figure 6).

The paper modeled the degradation feature with the extracellular features, mean area, and mean circularity.

The paper modeled the collagen degradation with respect to area, circularity and grayscale histogram mode.

Validity of the findings

In terms of the methodology, the pipeline of processing, analysis, modeling is general.

The time dependency model, the overall degradation model are properly modeled. Given the relatively small data set of 60 images, the 3-folder cross validation result is reasonable.

The findings by feature correlation analysis and modeling is novel.

Additional comments

The literature review and description needs to be revised.

The modeling and time dependency findings are interesting, application/practical usability of needs to be justified.

Reviewer 2 ·

Basic reporting

1, The English has to be improved to make it easier to understand. Some examples:
(1) Line 255-256: To achieve this goal, the classes of ROI were first defined. Based on the identification 
of these classes, a model is estimated from the training sample and. It associates the ROI of 
each pixel with its…
→ separate clearly into two setenses.
(2) Line 629: This methodology could be 
also useful to study de degradation of other scaffolds appart from collagen type I.
→ de degradation. appart -> apart 

(3) Line 728: Moreover, by 
tudying the parameters of the first and second logistic components of the model, 
…
→ tudying is a typo
(4) Line 48: It's not clear what the authors mean with 'since it is mainly of a qualitative nature'.
(5) Line 25, Line 148 and Line 187 sentences are hard to understand.

2, In both the article and the figures, the authors have to check carefully with names of the three cell groups (CCO, CCT and CO). Currently the mix-and-match is confusing.
(1) In the article I see four: CCO, COO (Line 498), CO, CCT.
(2) Line 642: Differences between fittings corresponding to CCT and CCT groups have
643 been found studying the first and second logistic functions, ...
→ CCT and ?

2, In the whole literature, although the authors mentioned multiple times, I failed to find a reference mentioning why collagen is a key scaffold to study bone regeneration. It'd add great value to this study, it the authors can find support information for (1) this is important, (2) this is similar to some other important biological questions that might use this method.

3, Although "a random forest classification model" has been mentioned in the abstract, introduction, and keywords, I failed to find details of the model is provided in the manuscript.
Line 169: Proposal of an automatic procedure, based on the combination of image segmentation techniques and supervised classification Random Forest, to identify the different regions of interest in the images obtained by optical microscopy. It is important to emphasize that the Random Forest classification model is trained taking into account the information pro- vided by qualified laboratory personnel in the qualitative study of this type of images. We present, therefore, a model that is capable of automating a complex and time consuming task in the context of this type of analysis.
→ However, no data/model/significance were provided for raw features definition by experts group or the random forest model. As a key part of the study, the authors have to put that part into the manuscript.

4, In the introduction part, (Line 61)the authors expressed their concerns on method based on human-decision. In Line 122-129, they mentioned multiple methods with concern that none can fit all cases. The authors have to make it more clear if their method can concur both points with method comparison, otherwise remove these points.

Experimental design

1, The authors provided here a case study. Without providing raw information of how their expert team selected features at the first place, and include in article/script how they built random forest tree model to select features, their results cannot be repeated.
Moreover, the authors didn't not provide information on whether their trained model work on other dataset of the same kind, or other biological question.

2, The authors raised the question of automatic and object method to measure bone regeneration in an in vitro system. However, without description of current method undertaken to fulfill the same target, or comparison with certain methods mentioned in the script, it's hard to evaluate how meaningful the question is.

Validity of the findings

The authors included only one dataset as their training set. Although a cross-validation has been provided in Figure9, it's not a comprehensive cross-validation as the features (the non-mentioned random forest tree model from experts provided features) were not selected only based on a subset of samples.
The authors didn't provide a golden-standard of mass loss in their experiments across CCO, CCT and CO. It'd be very helpful to evaluate their results if they can add these information.

Additional comments

The authors built a novel pipeline with public algorithms/packages for a case study, with (1) expert provide measurement of features; (2) use random forest tree to select feature from (1); (3) use the classifier on the four features to process image segmentation; (4) modeling with GAM and two-factor logistic model to look into biological insight of the experiment.

1, The authors mentioned one improvement of their method is to use classifier (machine learning) instead of Threshold to process segmentation. In Figure2 and 3, it's hard for human-eyes to directly judge which method is better, a more detailed comparison is required to support the authors' point. Other than that, please show the results if we feed the threshold-based segmentation into down-stream analysis, would it provide a significantly worse model in either GAM or logistic mixed model.

2, The authors mentioned other methods in Line 624, any possibility to provide a comparison?

3, Line 628: while the first logistic component accounts for the 50% of collagen mass, it'd be more careful to think the model can explain the degradation process by calling it "other degradation factor". As the authors also mentioned that 'experimental error' is included. More insight from the biological or medical side to interpret the output might be a huge plus of the understanding.

4, Fig 4 and 5 provide similar information. A log-transform of area and circularity values would help to show the pattern with other factors.

5, Figure 6 panel E and F share the same title, I believe F is the 'Comparison between the second logistic component of CCO and CCT treatments'. Please edit that.

---

## Round 0.2 · Minor Revisions

The prior reviewers were not able to re-review so I went through the response letter and found that most of the concerns have been properly addressed. The revised manuscript has been greatly improved.

As an application oriented study, I suggest the authors to highlight the advantage of automatic processing and accuracy over manual inspection. How the image noise will affect the performance.

I also suggest the authors to provide the details for the automatic systems. It's better to provide the software with some demo data.

---

## Round 0.3 · accepted · Accept

The authors addressed my concern and provide two videos that shows a tutorial explaining all the process of image analysis for identifying the different Regions of Interest (ROI), i.e. collagen, cellular matrix, nuclei. I suggest the acceptance of the manuscript now.